# IgaA negatively regulates the Rcs Phosphorelay via contact with the RcsD Phosphotransfer Protein

Erin A. Wall[¤], Nadim Majdalani, Susan Gottesman[ID]*

National Cancer Institute, Bethesda, Maryland, United States of America

¤ Current address: Food and Drug Administration, Silver Spring, Maryland, United States of America
* Gottesms@mail.nih.gov

## Abstract

Two-component systems and phosphorelays play central roles in the ability of bacteria to rapidly respond to changing environments. In *E. coli* and related enterobacteria, the complex Rcs phosphorelay is a critical player in the bacterial response to antimicrobial peptides, beta-lactam antibiotics, and other disruptions at the cell surface. The Rcs system is unusual in that an inner membrane protein, IgaA, is essential due to its negative regulation of the RcsC/RcsD/RcsB phosphorelay. While it is known that IgaA transduces signals from the outer membrane lipoprotein RcsF, how it interacts with the phosphorelay has remained unknown. Here we performed in vivo interaction assays and genetic dissection of the critical proteins and found that IgaA interacts with the phosphorelay protein RcsD, and that this interaction is necessary for regulation. Interactions between IgaA and RcsD within their respective periplasmic domains of these two proteins anchor repression of signaling. However, the signaling response depends on a second interaction between cytoplasmic loop 1 of IgaA and a truncated Per-Arndt-Sim (PAS-like) domain in RcsD. A single point mutation in the PAS-like domain increased interactions between the two proteins and blocked induction of the phosphorelay. IgaA may regulate RcsC, the histidine kinase that initiates phosphotransfer through the phosphorelay, indirectly, via its contacts with RcsD. Unlike RcsD, and unlike many other histidine kinases, the periplasmic domain of RcsC is dispensable for the response to signals that induce the Rcs phosphorelay system. The multiple contacts between IgaA and RcsD constitute a poised sensing system, preventing potentially toxic over-activation of this phosphorelay while enabling it to rapidly and quantitatively respond to signals.

## Author summary

The Rcs phosphorelay system plays a central role in allowing enterobacteria to sense and respond to antibiotics, host-produced antimicrobials, and interactions with surfaces. A unique negative regulator, IgaA, attenuates signaling from this pathway when it is not needed, but how IgaA controls the phosphorelay has been unclear. We define a set of

---

**Data Availability Statement:** All relevant data are within the manuscript and its Supporting Information files including the zipped supporting files for each figure.

**Funding:** Funding for this research was supported by the Intramural Research Program of the NIH, National Cancer Institute, Center for Cancer Research. EAW was supported by a PRAT Fi2 fellowship GM123943 from NIGMS. The funders had no role in study design, data collection and analysis, decision to publish, or preparation of the manuscript.

**Competing interests:** The authors have declared that no competing interests exist.

critical interactions between IgaA and the phosphotransfer protein RcsD, including a periplasmic contact between IgaA and RcsD that mediates a necessary inhibition of Rcs signaling. Inhibition is further modulated by regulated interactions between the cytoplasmic domains of each protein, providing a sensitive regulatory switch.

## Introduction

Bacteria must constantly monitor the integrity of their cell wall and envelope to withstand environmental insults. Osmotic stress, redox stress and envelope disruption demand that the bacterium remodel its exterior to provide protection, often via synthesis and secretion of capsular polysaccharide. Enterobacterales use the Rcs phosphorelay to integrate complex signals from the outer membrane and periplasm, changing gene regulation in response to stress [1, 2]. The Rcs phosphorelay is a complex signal transduction pathway, comprising an outer membrane lipoprotein (RcsF) and three inner membrane proteins (IgaA, RcsC and RcsD); these control the phosphorylation state and thus the activity of the transcriptional regulator (RcsB) (Fig 1A). RcsB in turn regulates production of virulence-associated capsules as well as motility and the expression of other stress-related genes.

Signaling through this pathway is not fully understood. It is known that outer membrane stress generated by cationic polypeptides or cell wall stresses produced by beta-lactams cause a change in the interaction between RcsF and IgaA. IgaA was originally identified in *Salmonella* and named for intracellular growth attenuation [3]; the *E. coli* homolog of this gene, *yrfF*, is referred to here as IgaA. The activated RcsF/IgaA interaction allows the hybrid histidine kinase RcsC to auto-phosphorylate and then pass phosphate to the phosphorelay protein RcsD, a process studied here, which then passes it to response regulator RcsB (Fig 1A). Over-signaling through the phosphorelay leads to cell death, possibly reflecting critical roles of multiple genes within the RcsB regulon. IgaA is essential due to its role as a gating/braking mechanism for the phosphorelay. Deletion of IgaA is only possible in cells that have mutations in *rcsC*, *rcsD*, or *rcsB* [4]. Multiple studies have focused on the interaction of RcsF with IgaA following cell wall stress [5–9], but little has been reported on the downstream action of IgaA [10]. In this work we define RcsD as the direct binding partner of IgaA and define the regions in RcsD that are critical for its interaction with IgaA. Production of RcsD variants that are deficient in IgaA binding cause massive over-signaling through the phosphorelay, resulting in mucoidy and poor viability, similar to phenotypes seen upon loss or inactivation of IgaA itself. Our findings support a model in which IgaA represses the phosphorelay through direct interaction with the phosphotransfer protein RcsD.

## Results

### A sensitive and flexible assay for the Rcs phosphorelay

We have reinvestigated the Rcs signaling pathway using a newly developed in vivo fluorescent reporter assay for expression of RprA, a small RNA that is a sensitive and specific target of Rcs regulation. RprA levels are nearly undetectable in the absence of RcsB, its direct transcriptional activator, and increase in cells in which the Rcs system is activated [11]. We constructed a transcriptional fusion of the *rprA* promoter to mCherry, which allows facile and continuous detection of Rcs activation over a wide range. This growth (S1A Fig) and fluorescence assay can be viewed as a function of fluorescence over average cell density, showing a rapid and significant change in slope in cultures treated with an Rcs stimulus (S1B and S1C Fig). Fig 1B shows a bar

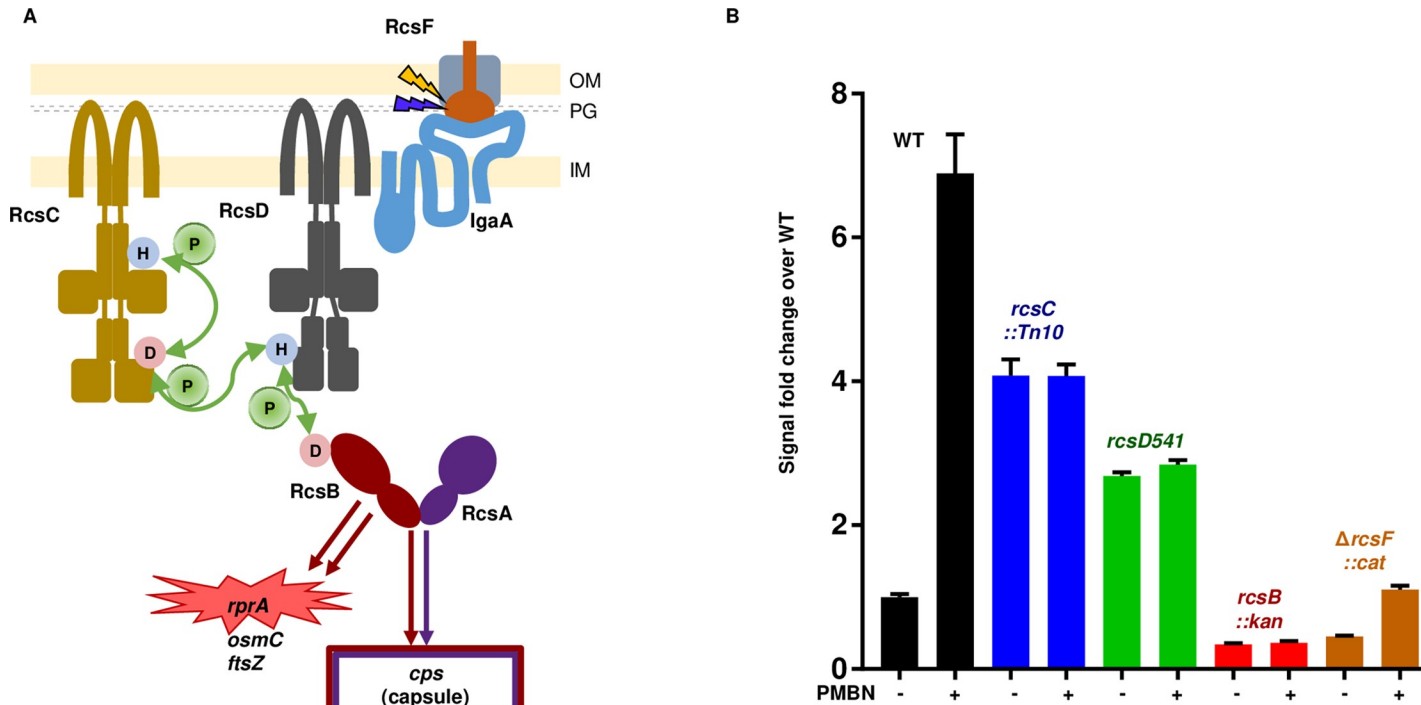

**Fig 1. Signaling via the Rcs Phosphorelay.** A. The six proteins of the Rcs Phosphorelay are shown schematically (not to scale; described in detail in [1]). RcsF (orange) is positioned in the outer membrane, associated with outer membrane porins (OMPs; grey). Most described treatments that induce the phosphorelay require RcsF for activation and thus it is shown as a key sensor for both outer membrane stress (represented by a gold lightning bolt) and periplasmic or peptidoglycan stress (dark blue lightning bolt). IgaA (blue) is a five-pass inner membrane protein that serves as a brake on the phosphorelay; it communicates with RcsF across the periplasm. Current models suggest that upon stress signaling, RcsF increases or changes contacts with IgaA, leading to de-repression of the phosphorelay. RcsC (gold) is induced to autophosphorylate and pass phosphate from its active site His 479 to its REC domain Asp 875. The phosphate is then passed to His 842 on the RcsD (dark grey) histidine phosphotransfer domain, and from there it passes to the RcsB (crimson) REC domain Asp 56. Phosphorylated RcsB forms homodimers or heterodimers with RcsA (purple) to regulate many genes. Shown here are induction of capsule synthesis by the RcsB/RcsA heterodimer and induction of the sRNA RprA by the RcsB homodimer. The red highlight around *rprA* indicates that an *rprA* promoter fusion to mCherry (P$_{rprA}$-mCherry) is used throughout this work to evaluate activation of the phosphorelay. Note that as with many phosphorelays of this family, phosphate can also flow in reverse from RcsB towards RcsC. IgaA is shown closest to RcsD, based on data presented in this study. In this schematic, RcsC and RcsD are depicted as homodimers, but their state is not currently known. B. The promoter of the sRNA RprA was fused to mCherry to create a reporter for Rcs activation (P$_{rprA}$-mCherry), that demonstrates sensitivity and a wide dynamic range. Activity of wild type cells (black, EAW8) was compared to *rcsC::Tn10* (blue, EAW18), *rcsD541* (green, EAW19), *rcsB::kan* (red, EAW31) and Δ*rcsF::cat* (orange, EAW32). All strains were also tested with polymyxin B nonapeptide (PMBN) at 20 μg/ml. Cells were grown in MOPS minimal glucose for the P$_{rprA}$-mCherry assay; signal shown is for cells at a density of OD$_{600}$ 0.4. Details of the assay and cell growth are shown in S1A, S1B and S1C Fig and described in Materials and methods.

graph of the fluorescence for cells at OD$_{600}$ 0.4 after wild-type and mutant cells are exposed to polymyxin B nonapeptide (PMBN), a non-toxic small molecule stimulator of Rcs signaling. PMBN stimulus is a useful indicator of pathway status. WT cells are induced in response to PMBN (compare WT bars in Fig 1B); pathway disruptions, by modification or deletion of pathway components, causes dampening or loss of the PMBN response. Cells with *rcsB* inactivated lose all signal, even the low basal level seen in the absence of PMBN (Fig 1B, S1A, S1B and S1C Fig). The absence of RcsF also lowers overall signal (compare Δ*rcsF::cat* to WT,— PMBN, Fig 1B). Decreased basal level signaling in the absence of RcsF has been reported before [5, 12–14], but is unambiguous with this assay. Lack of RcsF also greatly dampens the response to PMBN (Fig 1B). It is known that Rcs signaling can be induced in the absence of RcsF, and it is likely that the slight activation by PMBN observed with the Δ*rcsF::cat* strain occurs by this still unknown mechanism [12, 15]; (Majdalani et al, unpublished).

The hybrid histidine kinase RcsC and the phosphorelay protein RcsD play both positive and negative roles in regulation of RcsB activity. In the absence of RcsC or RcsD the response

to PMBN was blocked, but the levels of $P_{rprA}$-mCherry were significantly higher even in the absence of an inducing signal (Fig 1B). These findings are consistent with earlier work that expression of an $P_{rprA}$-*lac* reporter was increased upon deletion of *rcsC* or *rcsD* [11, 13]. Cells deleted for *rcsC* are thought to have lost the ability to de-phosphorylate RcsB that has been phosphorylated from other sources [16–18]. Under our assay conditions, the *rcsC* deletion strain produced a signal that is 3–4 fold above WT, and the *rcsD*541 mutant increased $P_{rprA}$-mCherry expression comparable to that observed with the *rcsC* deletion. Other *rcsD* alleles were also examined (see S1D and S1E Fig) and had reasonably consistent behavior.

The increased expression of $P_{rprA}$-mCherry seen in all *rcsD* and *rcsC* strains is likely due in part to phosphorylation of RcsB by the small molecule acetyl phosphate (AcP), combined with the lack of the dephosphorylation activity in the absence of RcsD and RcsC [17, 19]. The influence of AcP is easily seen in an *ackA* deletion strain that accumulates large intracellular pools of AcP [17] (S1F Fig). In an *ackA* mutant background, cells wild-type for *rcsD* and *rcsC* showed a modest increase in signal (compare black and gray bars), whereas the combination of an *ackA* mutant with mutations in *rcsC* or *rcsD* resulted in high levels of $P_{rprA}$-mCherry. The increase in signal was fully dependent upon RcsB, as expected (rightmost bar in graph, S1F Fig).

## IgaA and RcsD interact directly

We began interrogating how IgaA might interfere with Rcs signaling by examining the interactions of IgaA with downstream members of the phosphorelay, using the bacterial adenylate cyclase two hybrid assay (BACTH). In this assay, synthesis of beta-galactosidase in adenylate cyclase mutant cells is dependent on reconstitution of *Bordetella* adenylate cyclase from two fragments (T18 and T25, Cya). Each fragment is expressed from a separate vector as a fusion to one of a pair of potentially interacting proteins [20, 21]. The IgaA fusion interacted robustly with the RcsD fusion in two orientations (IgaA-T18/RcsD-T25 and IgaA-T25/RcsD-T18), and expressed beta-galactosidase activity approximately 20-fold greater than either fusion paired with an empty cognate vector, the standard background control (Fig 2A; S2A Fig). This interaction occurred irrespective of the chromosomal presence or absence of other Rcs members (S2B and S2C Fig). However, by the same assay, no significant interaction was detected between IgaA and RcsC (Fig 2A, S2B and S2D Fig). A derivative of RcsC missing the periplasmic region also failed to interact with either RcsD or IgaA (S2E Fig).

Cells expressing IgaA, RcsD, and RcsC fused to Cya fragments produced immunoreactive proteins of the expected size, and the fusion proteins were functional in the Rcs phosphorelay (see Materials and methods, S2F–S2I Fig). This suggests that the RcsC-T25 construct is not completely misfolded. However, we were unable to demonstrate the interaction of RcsC-T18, or of a derivative deleted for the periplasmic region of RcsC with itself or with RcsD (S2D and S2E Fig). Therefore, while we are confident of the interaction of IgaA with RcsD, we cannot interpret the lack of interaction of RcsC with IgaA as demonstrating that these never interact.

Regions in RcsD necessary and sufficient for interaction with IgaA were defined using the bacterial two-hybrid assay (Fig 2B). $RcsD_{1-383}$, a truncation that includes the trans-membrane and periplasmic regions but is missing most of the cytoplasmic regions including the incomplete Per-Arndt-Sim (PAS-like) domain, resulted in a loss of measurable IgaA interaction. However, IgaA interacted well with $RcsD_{1-461}$, a C-terminally truncated RcsD that includes the PAS-like cytoplasmic domain (Fig 2B). PAS domains are associated with signal detection in sensor histidine kinases [22]. Somewhat longer RcsD derivatives ($RcsD_{1-683}$, lacking the ABL and Hpt domains and $RcsD_{1-522}$, lacking the HATPase, ABL and Hpt domains), varied in their strength of interaction. It seems possible that the $RcsD_{1-522}$-T25 fusion derivative, with a

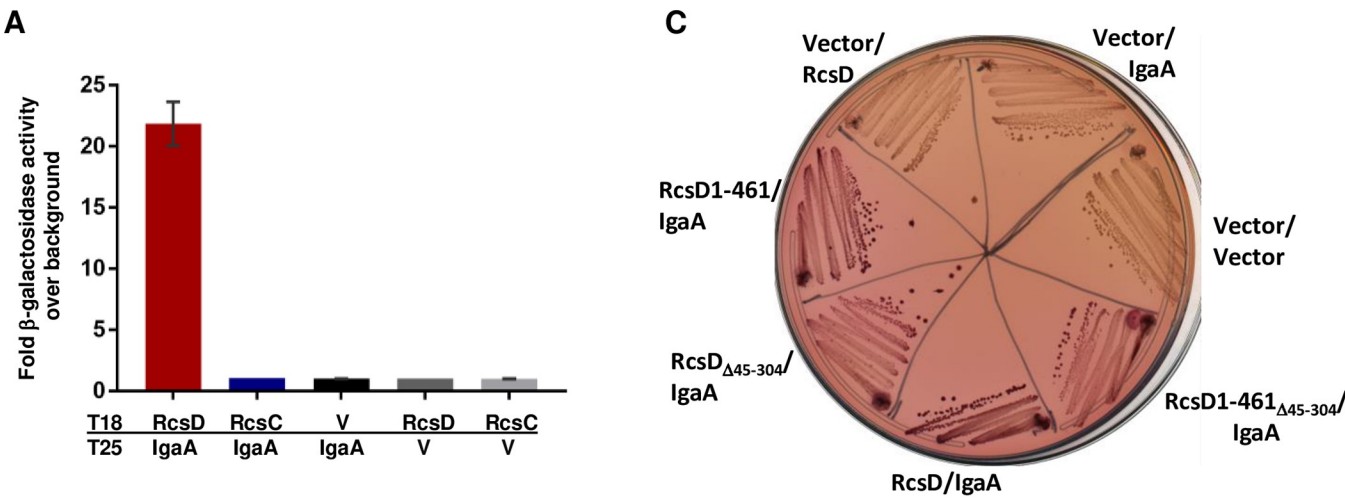

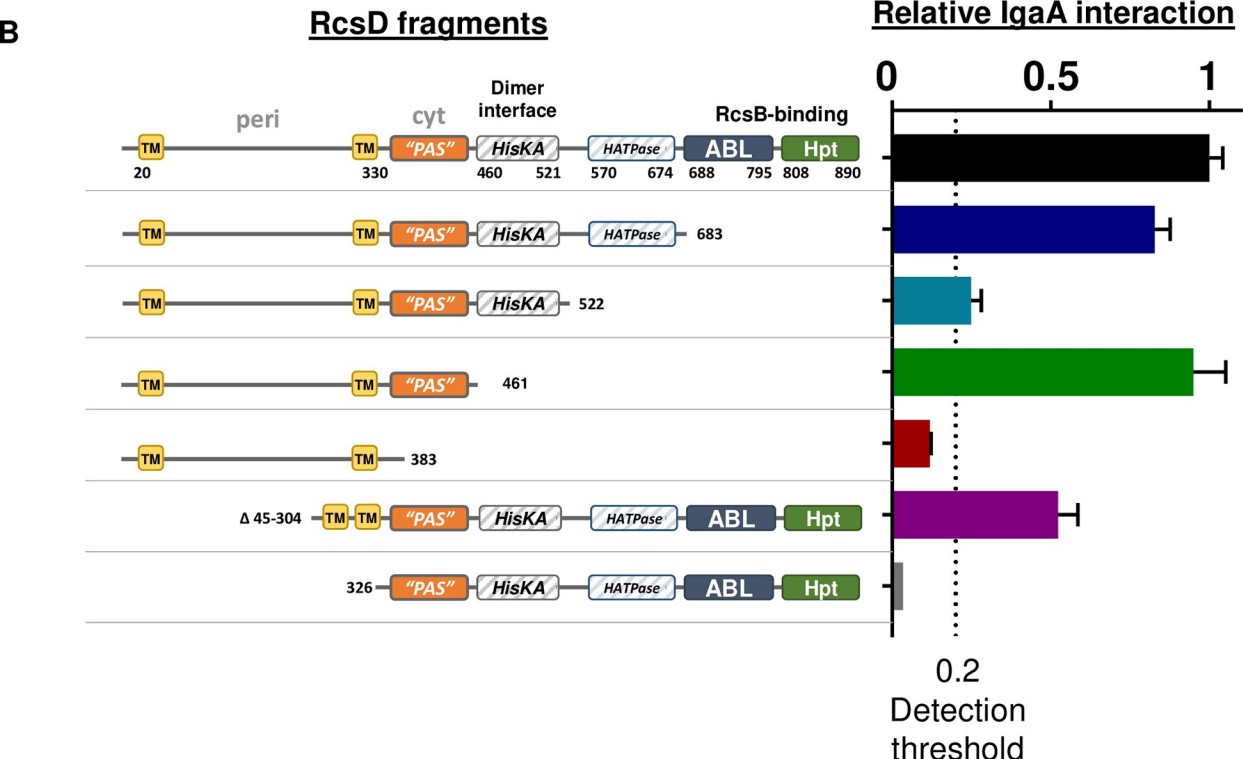

**Fig 2. Interaction of IgaA with RcsD.** A. Beta-galactosidase activity was measured in *cyaA* deficient cells (strain BTH101) containing a dual plasmid system encoding the T18 and T25 domains of adenylate cyclase fused to proteins of interest and the expression measured compared to background. Each protein fusion plasmid paired with its cognate vector (V) produces very little activity; the three controls were averaged and used as "background" for normalization. Error bars (some too small to be visible) represent standard deviation of three assays. Fusions present are IgaA-T25, RcsD-T18, and RcsC-T18. Beta-galactosidase activities, measured in Miller units, results obtained with the fusions in the opposite orientation (IgaA-T18, RcsD-T25, RcsC-T25), and in strains mutant for *rcs* genes to test roles of other Rcs proteins on the interaction, are shown in S2A–S2D Fig. B. Relative ratio of RcsD fragment binding to IgaA was determined by comparing the interaction of RcsD truncations to the interaction between full length RcsD and IgaA, normalized to 1 (top black bar). Extent of RcsD present is shown schematically next to the bar graph. The dotted line at y = 0.2 represents the threshold set here for reliable interaction detection, 4x over background signal. This data is compiled from separate sets of assays, each normalized relative to the IgaA/RcsD signal in that experiment. In most cases the IgaA/RcsD interaction is 20x over background, usually 1000 Miller units compared to 50 Miller units for the background control. All measurements were

carried out in strain BTH101. Plasmids used were pEAW1 (IgaA-T18), pEAW8 (RcsD-T25), pEAW8b (RcsD$_{1-683}$-T25), pEAW8α (RcsD$_{1-522}$-T25), pEAW8m2 (RcsD$_{1-461}$-T25), pEAW8m (RcsD$_{1-383}$-T25), pEAW8peri (RcsD$_{\Delta45-304}$-T25), and pEAW8s (RcsD$_{326-C}$-T25). C. Interactions of IgaA with RcsD derivatives expressing the PAS-like domain. A Lactose MacConkey plate with Ampicillin and Kanamycin was streaked with BTH101 co-transformed with T18 and T25 plasmids and incubated for two days at 30˚C. RcsD-T25 plasmids: RcsD (pEAW8), RcsD$_{1-461}$ (pEAW8m2), RcsD$_{\Delta45-304}$ (pEAW8peri), RcsD$_{1-461, \Delta45-304}$ (pEAW8m2peri); IgaA-T18 plasmid (pEAW1). Expression of the RcsD-T25 proteins is shown in S2J Fig. Positive interactions are red.

poorer but measurable interaction, may have a folding defect that interferes with the interaction with IgaA, although the T18 derivative is well expressed (S2G Fig). A fully cytoplasmic RcsD construct (RcsD$_{326-C}$) that lacked the N-terminal membrane binding regions and periplasmic loop of RcsD did not interact with IgaA, whereas RcsD$_{\Delta45-304}$, which lacks only the periplasmic region, still interacted with IgaA, although at about half the level seen with the WT (Fig 2B). The ability of RcsD$_{\Delta45-304}$ and RcsD$_{1-461}$ to interact with IgaA suggested that the PAS-like cytoplasmic domain of RcsD, between aa 383 and 461, plays a critical role in the interaction with IgaA. A final construct, RcsD$_{1-461, \Delta45-304}$, was created and tested. This construct, containing the PAS-like domain and localized to the membrane but devoid of the periplasmic loop of RcsD, was able to interact with IgaA as well (Fig 2C). The periplasmic loop of RcsD, while it may contribute to the interaction, is not sufficient to give a signal in the BACTH assay (Fig 2B).

These results are most consistent with RcsD interactions with IgaA within the PAS-like cytoplasmic regions bounded by residue 461. However, this region only gives a robust interaction in the context of the membrane-bound version of RcsD, not when part of a fully soluble RcsD protein (RcsD$_{326-C}$).

Intriguingly, some of the constructs that displayed significant interactions with IgaA (RcsD$_{1-522}$, RcsD$_{1-683}$ and RcsD$_{1-461}$, but not RcsD$_{1-383}$ and RcsD$_{326-C}$) also caused mucoidy in the cloning strain (Stellar *E. coli*, Clontech; wild-type for all genes of the Rcs phosphorelay), suggesting that overproduction of these constructs causes activation of the phosphorelay. This activating function was further examined, using the P$_{rprA}$-mCherry reporter as an assay.

## Titration of IgaA by overexpression of truncated RcsD

Our BACTH assays suggested that IgaA-dependent repression of Rcs was a consequence of direct interaction between IgaA and RcsD. If so, overproduction of the domains of RcsD capable of this interaction might titrate IgaA away from the chromosomally-encoded RcsD, leading to unregulated signaling through the Rcs phosphorelay. The RcsD fragments studied in the BACTH experiments were cloned without fusion proteins or tags under the control of the arabinose-inducible pBAD promoter. These plasmids were assayed in both *rcsD*$^+$ and *rcsD*541 strains containing the P$_{rprA}$-mCherry reporter fusion to monitor activation of the Rcs phosphorelay. In the absence of arabinose, RcsD is expressed from the pBAD promoter at a level sufficient to complement the *rcsD*541 mutation, turning down signaling that occurs in strains lacking RcsD (Fig 3A, lower panel; compare RcsD to V). Expression of constructs missing the C-terminal Hpt domain of RcsD (for instance, RcsD$_{1-383}$ and RcsD$_{1-461}$) did not inhibit this signaling (Fig 3A, lower panel).

In the *rcsD*$^+$ host, overproduction of RcsD fragments capable of interacting with IgaA (RcsD$_{1-683}$, RcsD$_{1-522}$ and RcsD$_{1-461}$) resulted in high expression of the P$_{rprA}$-mCherry fusion, while overproduction of RcsD$_{1-383}$, which did not interact with IgaA (Fig 2B), also did not stimulate the phosphorelay (Fig 3A, S3A Fig). The requirement for arabinose for the activation of the phosphorelay suggests that the RcsD interacting fragments need to be expressed at high levels to interfere with signaling by IgaA. Significant cellular growth arrest and lysis occurred when RcsD$_{1-683}$ or RcsD$_{1-522}$ were overproduced, making quantitative comparisons between

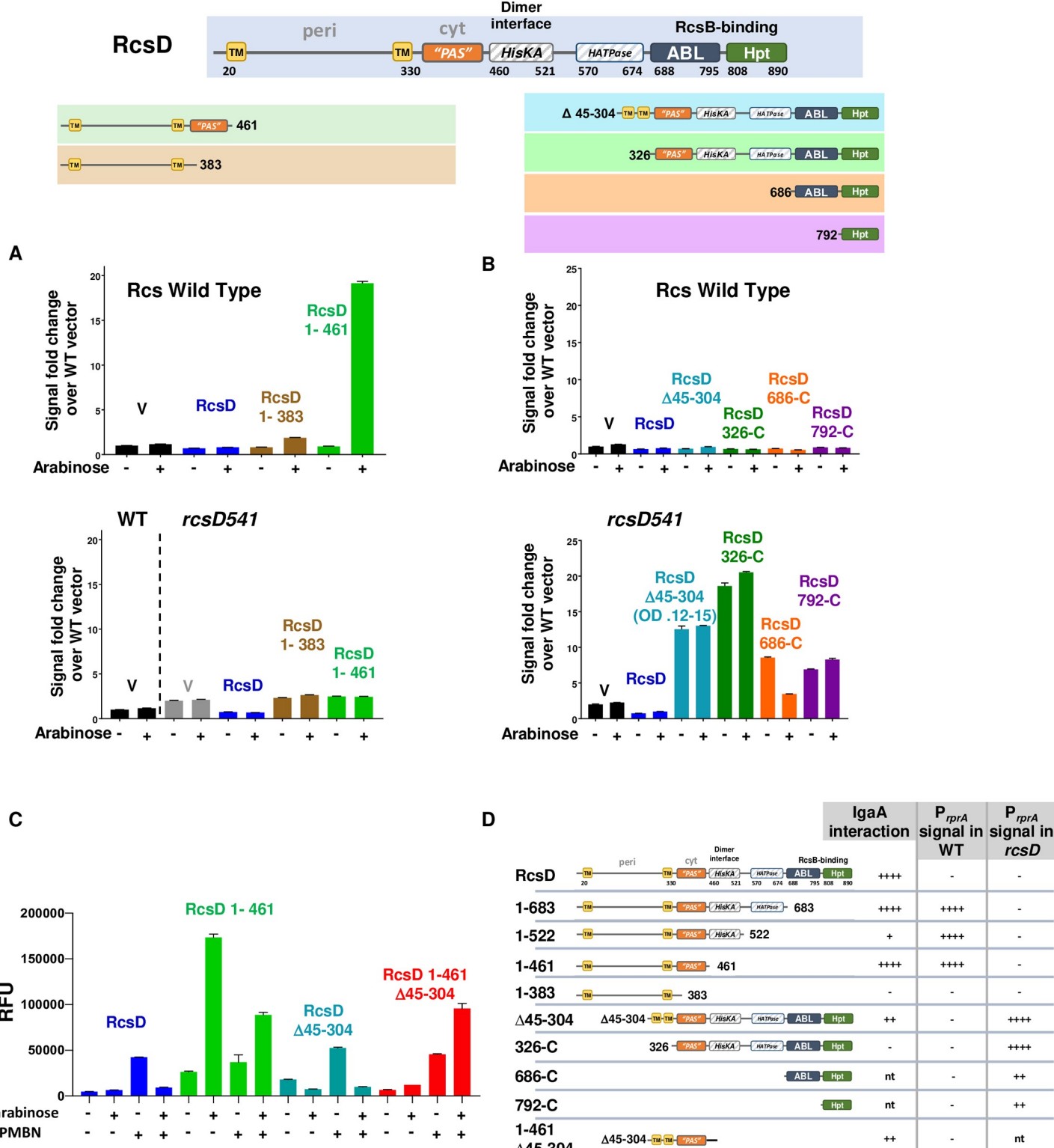

**Fig 3. Activity of truncated RcsD proteins.** A. RcsD wild type (upper bar graph, EAW8) and *rcsD541* mutant (lower bar graph, EAW19) cells carrying the P$_{rprA}$-mCherry fusion were transformed with pBAD24-derived plasmids encoding RcsD or C-terminally truncated pieces of RcsD, grown in MOPS 0.2% glucose with ampicillin (-arabinose) or MOPS glycerol with ampicillin with 0.02% arabinose (+ arabinose). Relative fluorescence values for each strain are shown at OD$_{600}$ 0.4, compared to the WT strain with vector. RcsD truncations used are shown at the top of the figure, with color-coding: black: V (vector, pBAD24); blue: RcsD (pEAW11); brown: RcsD$_{1-383}$ (pEAW11m); green: RcsD$_{1-461}$ (pEAW11m2). Note that the first two bars in the lower graph, to the left of the vertical dotted line, are in the *rcsD$^+$* host,

not *rcsD*541, to allow comparison of the *rcsD*+ and *rcsD*541 strains with the vector. Fluorescence as a function of $OD_{600}$ and additional related plasmids in the same strains are shown in S3A Fig; results in other strain backgrounds are shown in S3B and S3C Fig. B. Experiments as in panel A, but with plasmids carrying truncations of the N-terminus of RcsD, in *rcsD*+ (Rcs wild type; EAW8) and *rcsD*541 (EAW19) hosts. The constructs are color-coded as follows: black: vector (pBAD24), blue: full length RcsD (pEAW11), cyan: $RcsD_{\Delta45–304}$ (pEAW11peri), green: $RcsD_{326-C}$ (pEAW11s), orange: $RcsD_{686-C}$ (pEAW11c), purple: $RcsD_{792-C}$ (pEAW11d). Note that for the *rcsD*541 cells carrying $RcsD_{\Delta45–304}$, the value shown is at a low OD, the total achieved within 6 hours. C. Cultures of EAW8 (Rcs wild type, carrying the $P_{rprA}$-mCherry fusion) transformed with three of the plasmids tested in panels A and B, RcsD (blue), $RcsD_{\Delta45–304}$ (cyan), and $RcsD_{1-461}$ (green), as well as $RcsD_{1-461, \Delta45–304}$ (pEAW11m2peri, red) were grown under four conditions, with and without arabinose (as for A and B) and with and without PMBN (40μg/ml). Cultures grown without arabinose were grown in 0.2% glucose. D. Chart summarizing tests of RcsD fragments, their interaction with IgaA and ability to stimulate expression of the $P_{rprA}$-mCherry reporter in the absence of an inducing signal. Results from Figs 2B and 3A–3C and S3A Fig. nt: Not tested.

the constructs difficult. Nevertheless, the accumulation of PrprA-mCherry fluorescence even at low cell density (before lysis) and the significant activation of signaling were unambiguous (S3A Fig).

The ability of RcsD fragments to activate the phosphorelay in an *rcsD*+ host correlates well with their ability to interact with IgaA in the BACTH assay, consistent with the model that fragments interact with IgaA and free wild-type RcsD from IgaA repression. These results also reinforce the conclusion from the BACTH assays that a region, between aa 383 and 461, which includes the incomplete PAS-like domain, is required for the titration of IgaA.

Overexpressing full-length RcsD did not induce mucoidy or signaling. Since the intact RcsD can bind and potentially titrate IgaA, this result raised the possibility that one or more of the C-terminal domains of RcsD, absent in the activating/titrating truncated proteins, might exert an inhibitory effect on the phosphorelay. This is consistent with the report that the ABL domain binds RcsB and inhibits its phosphorylation [23].

The RcsD Hpt domain is known to be necessary for transmitting a signal from the RcsC response regulator domain to RcsB [24]. Therefore, we expected that plasmids expressing truncated RcsD constructs that lack the Hpt domain would be completely unable to activate the phosphorelay in *rcsD* mutants. In the *rcsD541* mutant allele background, the basal level of $P_{rprA}$-mCherry expression is higher than that in a WT host (Figs 1B and 3A, bottom panel, compare V to WT V). A plasmid expressing the intact RcsD reduced expression to levels comparable to the WT strain, consistent with complementation of the *rcsD*541 mutant (Fig 3A, bottom panel). The activating fragment $RcsD_{1-461}$ did not induce significant $P_{rprA}$-mCherry expression in this host, consistent with expectation, since it does not contain the Hpt domain (Fig 3A). Unexpectedly, cells expressing somewhat longer RcsD fragments, truncations $RcsD_{1-522}$ and $RcsD_{1-683}$, were able to significantly increase signal when induced in the *rcsD*541 host, even though both were missing the Hpt domain (S3A Fig, right-hand panel and S3B Fig). This is further discussed in S1 Text.

## Signaling by RcsD truncations with reduced capacity to interact with IgaA

Rcs signaling was also assayed in WT and *rcsD*541 strains carrying plasmids expressing various C-terminal portions of RcsD; all included the Hpt domain. Expression of the ABL-Hpt domains ($RcsD_{686-C}$), the Hpt domain alone ($RcsD_{792-C}$), the cytoplasmic portion of RcsD ($RcsD_{326-C}$), or RcsD deleted for the periplasmic region ($RcsD_{\Delta45–304}$) did not induce signaling in wild type cells (Fig 3B, upper graph). However, each of these plasmids led to significantly increased signaling in the *rcsD*541 strain, even in the absence of arabinose (Fig 3B, bottom graph). This high-level activity, particularly striking for $RcsD_{326-C}$ and $RcsD_{\Delta45–304}$, is best explained by these proteins sharing two critical properties–availability of an active Hpt domain to pass signal from RcsC to RcsB, and lack of effective repression by IgaA. As expected, the induction activity is dependent upon RcsC (S3D Fig).

$RcsD_{326-C}$ did not interact well with IgaA in the BACTH assay, although it contains the critical PAS-like domain. We suggest this is because it lacks membrane localization (Fig 2B). The

shorter $RcsD_{686-C}$ and $RcsD_{792-C}$ were not tested in the BACTH assay. $RcsD_{\Delta45-304}$, however, did interact, albeit at a somewhat reduced level. Cells expressing $RcsD_{\Delta45-304}$ grew poorly in the microtiter plates (6 hour $OD_{600}$ of 0.12, used in Fig 3B), and colonies containing this plasmid became mucoid in the absence of arabinose induction. Therefore, while IgaA is able to interact with RcsD deleted for the periplasmic region, we suggest that effective repression requires that RcsD include the periplasmic domain.

This was further tested by comparing the behavior of overproduced $RcsD_{1-461, \Delta45-304}$ in a $rcsD^+$ host to the behavior of RcsD, $RcsD_{1-461}$, and $RcsD_{\Delta45-304}$. Cells containing the pBAD-RcsD plasmids were grown with and without arabinose induction of the pBAD promoter, as well as with and without PMBN (Fig 3C). The proteins were all expressed (S4A Fig). Overexpression of intact RcsD had no effect on Rcs signaling on its own, but inhibited the ability of PMBN to induce signalling, consistent with evidence that the ABL-Hpt domains can inhibit RcsB [23]. Overproduction of $RcsD_{1-461}$, as in Fig 2A, effectively titrated IgaA; this was also reflected in a significant effect on cell growth (S4B Fig, right panel). Overproduction of $RcsD_{\Delta45-304}$ did not titrate, as noted above, and, presumably because it does have the ABL-Hpt domains, interfered with PMBN induction of the chromosomal copy of RcsD (Fig 3C). $RcsD_{1-461,\Delta45-304}$, expressing the PAS-like domain in the context of a membrane-localized protein, was unable to titrate (no activity with arabinose alone). The comparison of $RcsD_{1-461,\Delta45-304}$ to $RcsD_{1-461}$ supports a requirement for the periplasmic domain of RcsD for titration. Intriguingly, however, $RcsD_{1-461,\Delta45-304}$ collaborated with PMBN, so that cells showed an increased signal with arabinose and PMBN compared to PMBN alone (Fig 3C). Growth of both $RcsD_{\Delta45-304}$ and $RcsD_{1-461,\Delta45-304}$, but not RcsD or $RcsD_{1-461}$ was slowed when both PMBN and arabinose were present compared to arabinose alone (right hand panel, S4B Fig, compare cyan and red curves to black and green curves); PMBN had very little effect on growth in the absence of arabinose (left hand panel, S4B Fig).

Selected alleles of the truncated RcsD proteins were introduced into the bacterial chromosome in place of the native *rcsD* gene and their behavior examined. In these strains, RcsD variants should be expressed from the native promoter, presumably at the physiological level. Alleles were introduced in parallel into either a strain carrying an *rcsB* deletion, where phosphorelay signaling is off, or into a wild-type host. All tested alleles could be introduced into the *rcsB* mutant strain, but some alleles were difficult to isolate or were clearly unstable in *rcsB*⁺ cells. *rcsD*$_{\Delta45-304}$ could not be constructed in the *rcsB*⁺ host without accumulating secondary loss-of-function mutations elsewhere in *rcsD* or in *rcsB*. A mutant that deleted slightly less of the periplasmic domain, *rcsD*$_{\Delta48-304}$, could be introduced into the chromosome, but the cells were mucoid and constitutively activated for the Rcs phosphorelay (S4C and S4D Fig). The *rcsD*$_{\Delta48-304}$ allele may be slightly defective for phosphorelay function, and therefore is better tolerated than the *rcsD*$_{\Delta45-304}$ allele. Cells containing *rcsD*$_{326-C}$ were also quite mucoid and had significant PMBN-independent signaling; RcsD ABL-Hpt was better tolerated (S4C and S4D Fig). The behavior of these mutants parallels the behavior of the corresponding RcsD plasmids in an *rcsD*541 strain (Fig 3B). The *rcsD* truncations, in addition to expressing higher levels of $P_{rprA}$-mCherry, did not respond to PMBN induction (S4C Fig).

Strains carrying the chromosomal *rcsD* mutations were tested for their ability to tolerate deletion of *igaA*. The igaA mutation was introduced into the recipient strain by P1 transduction from a donor (EAW66) containing a *bioH*::kan^R mutation closely linked to *igaA*::chl^R, in a strain containing an unlinked *rcsD*541 mutation. Kanamycin resistant transductants were isolated and tested for the closely linked chloramphenicol resistance marker in *igaA*::chl^R (S4D Fig). In recipient strains defective for *rcsB*, rcsC, or *rcsD*, the linkage of the *bioH*::kan and *igaA*::chl^R markers was >70%; in a strain WT for the Rcs phosphorelay, linkage was <1%, reflecting the known lethality of *igaA* mutation in cells with a functional Rcs phosphorelay.

Strains carrying the *rcsD*541 or *rcsD*841* mutations tolerated loss of *igaA* well, as expected for strains null for *rcsD* (S4D Fig). Also as expected, a strain deleted for *rcsB* (EAW54, S4D Fig) allowed introduction of the *igaA*::chl$^R$ mutation.

The *rcsD*$_{326-C}$ strain (EAW53), carrying all of the cytoplasmic regions of RcsD, did not tolerate loss of IgaA (S4D Fig). Although chl$^R$ colonies were isolated, those colonies had unstable phenotypes; restreaking yielded colonies that were less mucoid or fluorescent, strongly suggesting that cells carrying the *igaA* deletions could only survive when the Rcs phosphorelay was defective. These results support a model of a critical regulatory contact between IgaA and RcsD in the PAS-like domain. EAW106, expressing RcsD deleted for the periplasmic region (*rcsD*$_{\Delta48-304}$), also did not tolerate introduction of the *igaA* deletion (EAW106, S4D Fig).

A strain carrying *rcsD*$_{686-C}$ (ABL-Hpt), which encodes RcsD lacking all regions involved in IgaA interaction, displayed a lower level of signaling (S4C Fig), was non-mucoid, and, as expected, tolerated the loss of *igaA* (EAW108, S4D Fig). We would suggest, based on its phenotypes, that this construct is not fully active for passing signal from RcsC, via RcsD, to RcsB.

Our findings, summarized in Fig 3D, allow several conclusions. First, the RcsD periplasmic region is essential for repression by IgaA, but is not sufficient for binding to IgaA by the BACTH assay or for titration of IgaA. Our data are best explained by a direct interaction between the RcsD periplasmic loop and IgaA. The precise role of the trans-membrane (TM) regions flanking the periplasmic loop have not yet been explored. Constructs that lack the periplasmic loop but carry the Hpt domain are capable of induction-independent signaling, presumably because they are at least partially blind to IgaA repression. Second, a critical region of interaction with IgaA lies within the cytoplasmic PAS-like domain of RcsD. This PAS-like domain, in the context of a membrane-bound protein, is sufficient for interaction with IgaA. An RcsD protein carrying both the periplasmic interaction site as well as the PAS-like domain is sufficient to titrate IgaA; neither is sufficient on its own. The combination of these two interaction sites leads to robust repression in the absence of signal. Third, the Hpt domain on its own, or the full cytoplasmic domain, is recessive to RcsD$^+$, and is thus not able to constitutively signal in the presence of functional RcsD.

## Critical residues in the cytoplasmic PAS-like domain of RcsD

Alanine scanning mutagenesis of individual conserved residues in the PAS-like domain of RcsD was carried out in the pBAD-RcsD plasmid to look for mutations that affected interaction with IgaA. The pBAD-RcsD mutant plasmids were initially screened based on the fluorescence in an *rcsD*541 mutant strain grown in the absence of arabinose. In this assay, functional RcsD expressed from the plasmid reduces fluorescence by complementing the *rcsD*541 allele. A null mutation in the plasmid-borne RcsD copy would not affect fluorescence, and mutant RcsD alleles capable of passing phosphate from RcsC to RcsB that are less sensitive to IgaA repression were expected to have higher fluorescence (see lower panel in Fig 3B, for example). Unexpectedly, many of the plasmids that gave strong signals and were thus thought to be blind to IgaA instead had acquired stop codons within the *rcsD* open reading frame. These mutants were not further investigated. We instead focused on alanine mutants that retained RcsD function, measured by the ability to complement the *rcsD*541 mutant, reducing the elevated signal found in *rcsD*541 to the level found in *rcsD*$^+$ strains (compare lane 1 and lane 3, S5A Fig). Among the five mutants screened, one was unresponsive to PMBN induction. This mutant, *rcsD*T411A, was selected for further study because it behaved as though the mutant RcsD was somehow locked in an "off" configuration.

The *rcsD* T411A mutation was introduced into the chromosome and the strain was tested for its response to PMBN (Fig 4A). The mutant had a lower basal level of Rcs signaling and, as

was seen with the plasmid-borne copy, this mutant had a very muted response to PMBN. To determine if the lack of response was unique to PMBN, we tested the response of T411A to A22, an inhibitor of MreB, and to the peptidoglycan inhibitor mecillinam, both of which have also been reported to induce the Rcs System [5, 25]. Both compounds induced our $P_{rprA}$-mCherry reporter (Fig 4A, WT). As with PMBN, the T411A mutant also failed to respond to A22 or mecillinam (Fig 4A, red bars).

We considered two ways in which T411A might block induction. It might affect the ability of the phosphorelay signal to move from RcsC through the phosphorelay to RcsB, possibly by locking RcsD in the "phosphatase" conformation. Such a defect would be expected to make RcsDT411A insensitive to IgaA. Alternatively, $rcsD$T411A might be locked off because it cannot be released from IgaA when a stress signal is received. In this case, cells with the $rcsD$T411A mutation would, like $rcsD^+$, be sensitive to loss of IgaA. This is what we found. The strain carrying a chromosomal $rcsD$T411A mutation was not able to tolerate deletion of $igaA$ (EAW121, S4D Fig), suggesting that RcsD T411A cannot be activated because of increased or changed interaction with IgaA.

We next asked what regions of IgaA are likely to interact with RcsD and with RcsDT411. Based on our observations with the RcsD truncations (Fig 3D), we predicted that there should be interactions between the periplasmic domains of both proteins, as well as between their cytoplasmic domains via the PAS-like domain of RcsD.

The periplasmic domain of IgaA (shown schematically in Fig 4B) was previously found to interact with RcsF [10]. Here, we found that it is also likely to interact with RcsD, as predicted from our analysis of RcsD domains. In the BACTH assay, deletion of the IgaA periplasmic domain ($IgaA_{\Delta384-649}$) led to loss of the interaction of RcsD and IgaA. Interestingly, RcsD T411A restored some interaction, consistent with our expectation for a contact between the cytoplasmic domains of IgaA and RcsD, increased by the T411A (Fig 4B and S5B Fig). However, deletion of either cytoplasmic loop 1 or cytoplasmic loop 2 of IgaA had essentially no effect on the interaction with wild-type RcsD in the BACTH assay (Fig 4B), suggesting that the primary interactions detected by this assay are between the periplasmic regions.

We also examined the interaction of RcsD with IgaA carrying the periplasmic point mutation L643P, encoding a stable protein (S5C Fig) that caused a partial loss of function mutant in $igaA$ [26]. This mutation led to loss of interaction of IgaA and RcsD (S4D Fig). However, because other alleles at this position (L643A) or nearby mutations in highly conserved residues did not disrupt interaction or activity (S5D and S5E Fig), we conclude that L643 is not likely to directly participate in IgaA function or directly interact with RcsD. The introduction of proline in place of L643 may affect folding or localization of a critical region of IgaA, disrupting its ability to properly interact with RcsD.

Mutations that deleted one of the cytoplasmic loops (Δ36–181, cyt1 or Δ263–330, cyt2) or the periplasmic loop (Δ384–649, peri) of IgaA (see Fig 4B) were introduced in place of the chromosomal $igaA$ gene, in $rcsD541$ cells carrying the $P_{rprA}$-mCherry reporter. A mutant with a complete $igaA$ deletion was used for comparison. Introduction of the RcsD plasmid, even in the presence of glucose to maintain low levels of RcsD expression, was poorly tolerated in all the partial $igaA$ deletions as well as the complete deletion. Because secondary mutations arose at a rapid rate (see inset, Fig 4C), assays in liquid were considered untrustworthy, and the phenotypes of the primary transformants were instead evaluated on agar plates (Fig 4C).

Transformation of the RcsD plasmid into cells carrying a deletion of $igaA$, the $igaA$ periplasmic domain ($igaA_{\Delta384-649}$) or the second cytoplasmic domain ($igaA_{\Delta263-330}$) gave rise to highly mucoid growth, consistent with lack of IgaA function in attenuating Rcs expression. When the RcsD$^+$ plasmid was introduced into cells deleted for the first cytoplasmic domain of IgaA ($igaA_{\Delta36-181}$), streaked colonies were less mucoid, although the $P_{rprA}$-mCherry reporter

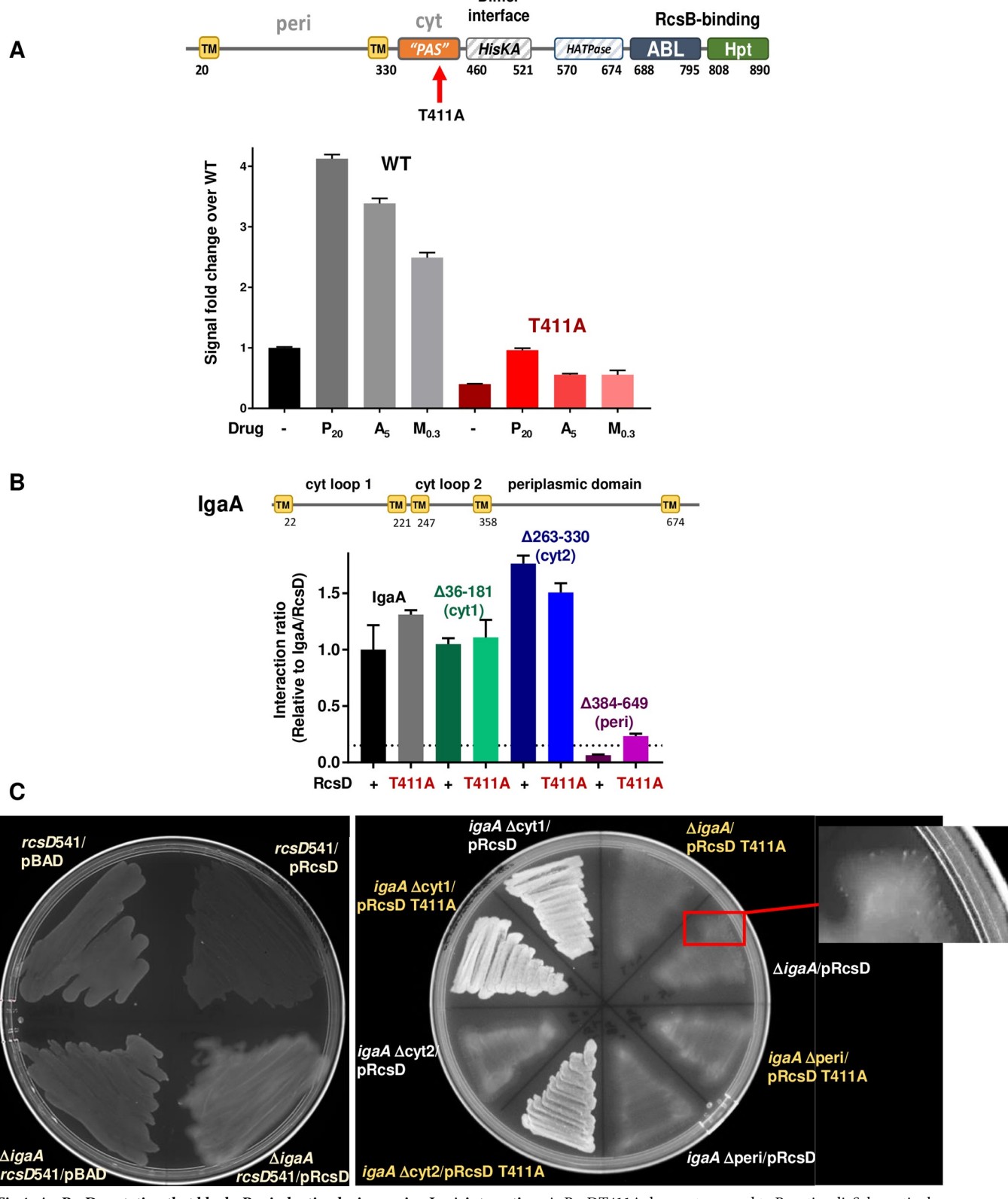

**Fig 4. An RcsD mutation that blocks Rcs induction by increasing IgaA interaction.** A. RcsDT411A does not respond to Rcs stimuli. Schematic shows domains of RcsD and position of T411A, within the PAS-like domain. Both wild type and *rcsD*T411A strains (EAW8 and EAW121) were treated with nothing (-), 20 μg/ml polymyxin B nonapeptide (P$_{20}$), 5μg/ml MreB inhibitor A22 (A$_5$) or 0.3μg/ml Mecillinam (M$_{0.3}$). Both A22 and Mecillinam give a smaller

dynamic range of wild type signaling than PMBN. B. RcsD PAS-like domain mutation T411A interaction with IgaA missing the periplasmic domain. IgaA schematic includes yellow transmembrane domains (TM), amino acid numbering, and loops labeled with their localization. BACTH results are shown as ratios relative to the wild type IgaA/RcsD interaction, which gave 1743 Miller units in this experiment. Plasmids used: IgaA-T18 (pEAW1); IgaA$_{\Delta36\text{-}181}$-T18 (pEAWcyt1); IgaA$_{\Delta263\text{-}330}$-T18 (pEAW1cyt2); IgaA$_{\Delta384\text{-}649}$ (pEAW1peri); RcsD-T25 WT (pEAW8) and RcsD T411A (pEAW8T). Background controls and fold above background values are shown in S5B Fig. C. RcsD and RcsD T411A plasmids in cells carrying the *rcsD*541 mutation and chromosomal *igaA* deletions. Strains were plated directly after transformation on MOPS ampicillin glucose plates and incubated overnight at 37˚C. Rcs⁺ strains devoid of IgaA activity are unstable and thus cannot be purified or assayed in liquid culture. The left plate contains control strains (clockwise from top left quadrant) *rcsD*541 with pBAD vector, showing moderate level of fluorescence as expected for a *rcsD* mutant (EAW19 with pBAD24), rcsD541 with RcsD⁺ on a plasmid, showing low level of fluorescence for a complemented (wild-type Rcs) strain (EAW19 with pEAW11), a *rcsD*541 Δ*igaA* strain with the RcsD⁺ plasmid, showing very mucoid growth associated with loss of IgaA; these cells generally will not form colonies on restreaking (EAW95 with pEAW11) and *rcsD*541 Δ*igaA* with pBAD vector, showing the same moderate fluorescence in the absence of RcsD (EAW95 with pBAD24). Note that mucoidy scatters the mCherry fluorescence, making it appear lower than the actual output. Right panel and inset, strains carrying indicated *igaA* deletions in the chromosome in an *rcsD*541 background, transformed with plasmids expressing either RcsD T411A (pEAW11T) or RcsD⁺ (pEAW11). The inset shows bright streaks within EAW95+pEAW11; this mucoid primary transformant spontaneously generates non-mucoid *rcs* mutants. Many of these mutants are not nulls, and the loss of mucoidy increases the apparent fluorescence. Therefore, these show up as more brightly fluorescent spots within the mucoidy. The brightly fluorescent sectors on this plate are bright because they are non-mucoid, reflecting a significant decrease in Rcs signaling but are still signaling at a level that is clearly much higher than in the dark, non-mucoid controls on the left-hand plate. Strains used, clockwise from top, two sectors for each strain, first with RcsDT411A, second with RcsD⁺: *rcsD*541 Δ*igaA* (EAW95); *rcsD*541 igaAΔperi (EAW98); *rcsD*541 igaAΔcyt2 (EAW97); *rcsD*541 igaAΔcyt1 (EAW96).

was well expressed compared to cells with WT IgaA (Fig 4C left panel). This phenotype was similar when a RcsDT411A plasmid was introduced. Therefore, deletion of cytoplasmic loop one was the best tolerated of the three IgaA partial deletions, but still plays an important role for proper IgaA repression of Rcs signaling. The plasmid expressing RcsDT411A was also not tolerated in strains containing either the full deletion of *igaA* or the deletion of the IgaA periplasmic domain. However, when the RscDT411A plasmid was put into cells carrying a deletion of the second cytoplasmic domain of IgaA, the colonies had reduced mucoidy and grew more robustly (Fig 4C). These results suggest that RcsD with the T411A mutation has improved interaction with IgaA cytoplasmic loop 1, allowing the mutated IgaA to regain partial function as a brake on RcsD signaling.

## Analysis of RcsC domains and involvement in signaling

From previous work, it is clear that RcsC plays an essential role in signaling in the Rcs phosphorelay, as the initiating kinase for the phosphorylation cascade [18]. However, as shown above, full length RcsC and RcsC deleted for the periplasmic domain did not interact with IgaA in the bacterial two hybrid assay (Fig 2 and S2 Fig). The plasmids expressing T18 and T25 fusions to full-length RcsC interfered with cell growth, and the fusion proteins did not interact with RcsD, and thus while our data strongly supports the interaction of IgaA with RcsD, we are cautious in interpreting this negative result with RcsC and IgaA.

Many RcsC constructs in pBAD24 were cytotoxic, causing massive cell lysis without any detectable increase in P$_{rprA}$-mCherry signal above background, and slow cell growth even in rich defined glucose media, where the pBAD promoter should be only modestly active. To avoid this toxicity, we introduced deletions and substitutions of interest into the chromosomal copy of *rcsC* and tested the response to induction by PMBN treatment using the P$_{rprA}$-mCherry reporter. As expected, cells carrying *rcsC*H479A, mutant in the kinase active site, had low expression and were not responsive to PMBN (Fig 5A). Note that the low activity in this mutant is more like that in the wild-type strain without PMBN than like the deletion of *rcsC* (Fig 5A), consistent with RcsCH479A retaining phosphatase activity, as reported by Clarke et al [18].

Unexpectedly, cells carrying *rcsC* $_{\Delta48-314}$, a mutation that deleted the periplasmic portion of RcsC, leaving the TM helices, responded strongly to PMBN, Mecillinam and A22 (Fig 5). PMBN induction in cells carrying *rcsC* $_{\Delta48-314}$ still required RcsF, strongly suggesting that the signaling pathway is not perturbed (Fig 5A). However, the *igaA* deletion could be introduced

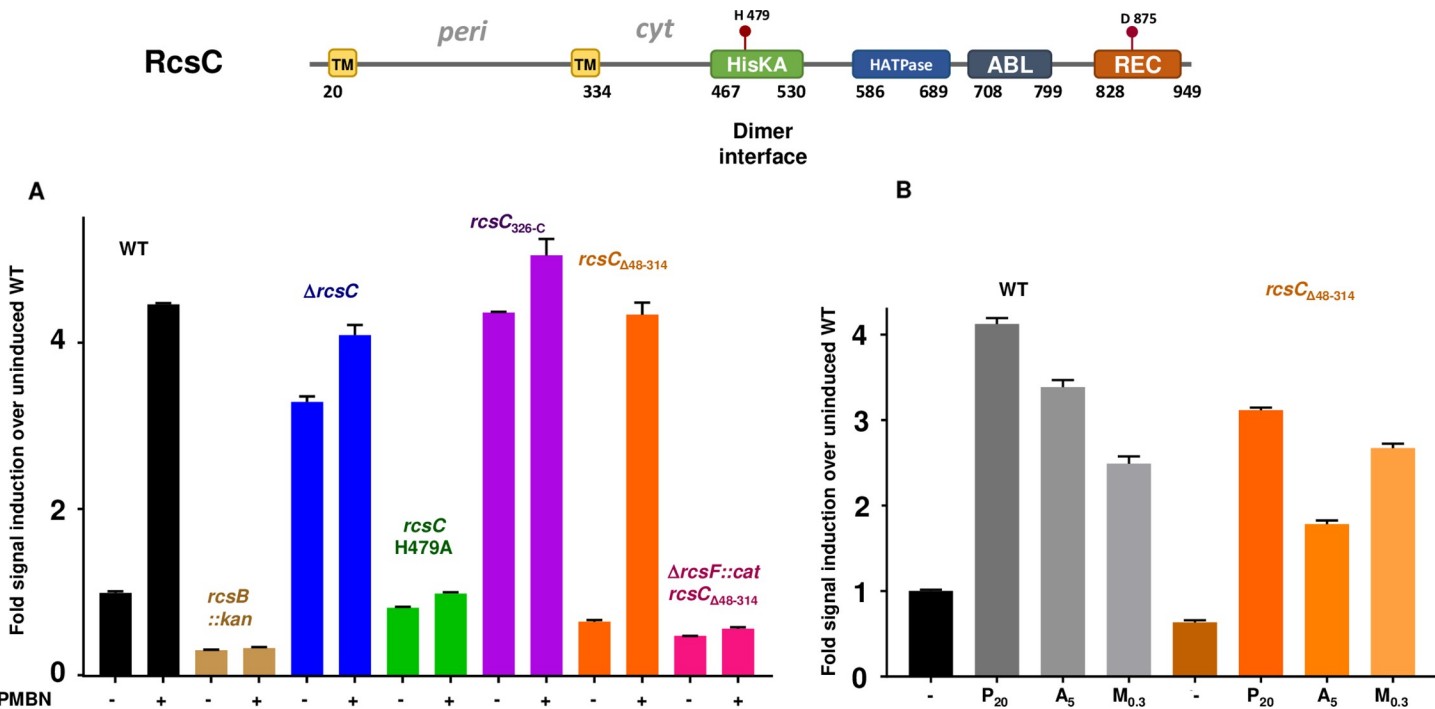

**Fig 5. The RcsC periplasmic region is dispensable for polymyxin B nonapeptide (PMBN), A22, and mecillinam-induced signaling.** The top panel shows a schematic of RcsC with domains, topology and active site residues noted. A. PMBN induction in various *rcsC* mutations. Strains included are (L to R) WT: EAW8, *rcsB::kan*: EAW31, Δ*rcsC*: EAW91, *rcs*CH479A, mutant in the active site histidine: EAW92, *rcsC*$_{326-C}$: EAW56, *rcsC*$_{Δ48–314}$: EAW70 and *rcsF::cat rcsC*$_{Δ48–314}$: EAW85. B. The effect of three Rcs stimulating drugs, PMBN, A22 and mecillinam (P$_{20}$, A$_5$, M$_{0.3}$) on WT and RcsC$_{Δ48–314}$. The RcsC periplasmic deletion strain has a lower basal level of signal than WT here; this was also seen with other RcsC periplasmic deletions (S6B Fig).

into cells carrying the *rcsC*$_{Δ48–314}$ mutation, although they became mucoid and unstable (EAW70, S4D Fig). Strains with a deletion of *rcsC* or with the *rcs*CH479A allele were unaffected when *igaA* was deleted (S4D Fig). We suggest that the *rcsC*$_{Δ48–314}$ allele likely has a modestly decreased ability to signal (decreased kinase activity and/or increased phosphatase activity) and/or to pass signal to RcsD, compared to the wild-type protein. Strikingly, our data rule out a role for the RcsC periplasmic loop both in mediating IgaA regulation of the phosphorelay and in signal transduction via RcsF.

Although the periplasmic region is not necessary for RcsC function, it would appear that membrane association is important. Cells carrying a deletion of the membrane spanning portion (RcsC$_{326-C}$) acted in a similar manner to an *rcsC* deletion, showing a constitutive level of reporter expression and no response to PMBN (Fig 5A). Consistent with a loss of function for the *rcsC*$_{326-C}$ allele, the deletion of *igaA* could be introduced into this strain, and cells remained non-mucoid (EAW56, S4D Fig). While this soluble portion of RcsC did interact with RcsD, it did not interact with IgaA (S6A Fig). A series of periplasmic deletions with different linker lengths all responded to PMBN to some extent (S6B Fig). Finally, a chimeric construct in which the MalF TM and periplasmic region replaced the *rcsC* periplasmic region also restored the ability of the cell to respond to PMBN (S6B Fig), albeit with a higher basal level of signaling in the absence of PMBN.

## Discussion

The results reported here provide a new view of how IgaA transduces inducing signals within the complex Rcs phosphorelay (Fig 1A, Fig 6). IgaA, a multipass membrane protein, is a strong

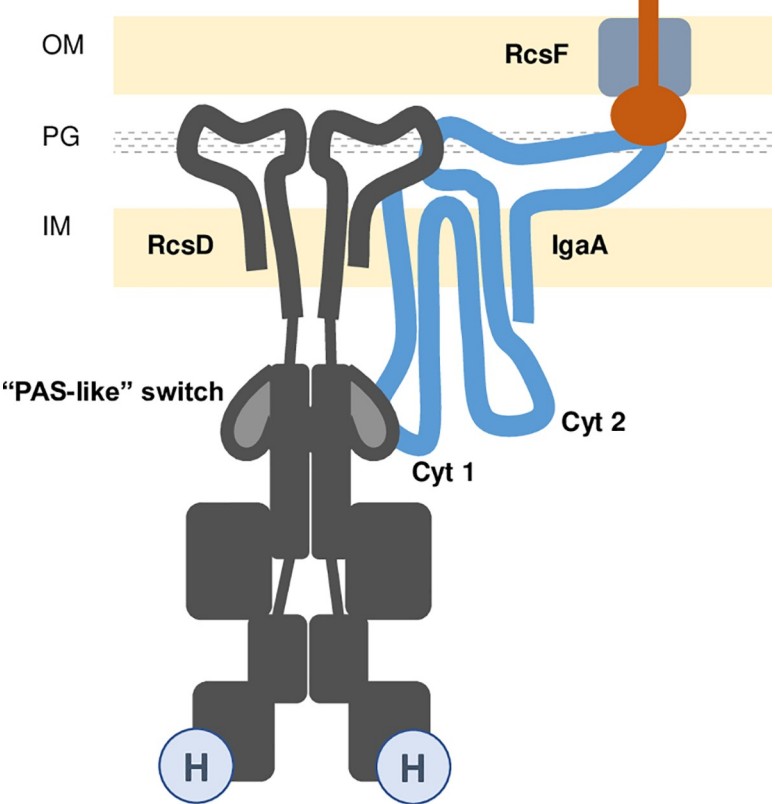

**Fig 6. Proposed interactions of IgaA and RcsD.** In this model, extensive interactions in the periplasm and in the cytoplasm are shown between IgaA and RcsD, consistent with genetic data indicating that signals pass from one compartment to the other via IgaA. RcsF is shown as in Fig 1A, poised to sense outer membrane disturbance and available to contact IgaA to change the course of signaling. Anchoring interactions between RcsD and IgaA in the periplasm contribute to the BACTH interaction signal and are required for IgaA repression of signaling. Interaction of the IgaA cytoplasmic loop 1 (Cyt1) and the PAS-like domain of RcsD (shown in lighter grey) are suggested to comprise the signal-switching interaction, tightened in *rcsD*T411A, an allele that blocks induction. Some aspects of this model are not known. It is unknown if RcsD acts as a dimer as shown; it is also unknown if RcsF and RcsD interact with separate (as shown) or overlapping parts of the periplasmic loop of IgaA.

negative regulator of Rcs. As previously described, inducers such as PMBN, believed to disrupt LPS interactions, A22, an inhibitor of MreB, and mecillinam, a beta-lactam antibiotic affecting peptidoglycan, act by changing the nature of the RcsF/IgaA interaction. This leads to a change, presumably a decrease, in IgaA's interaction with components of the downstream phosphorelay. We find that IgaA has critical interactions with the phosphotransfer protein RcsD. We were unable to detect interactions of IgaA with the RcsC histidine kinase. While this failure to detect a RcsC-IgaA interaction may reflect limits in the BACTH assays used, we also find that the TM and periplasmic portions of RcsC are not required for sensing and transducing the PMBN inducing signal (Fig 5, S6B Fig). In our model, the change in the IgaA-RcsD interaction allows RcsC-generated phosphate to flow from RcsC to RcsD, and from there to RcsB, activating RcsB-dependent transcription. Deletion and mutation analysis of RcsD identified multiple regions important for IgaA-dependent regulation, separate from the Hpt domain needed for phosphate flow from RcsC to RcsD and from RcsD to RcsB. These observations help to explain why RcsD includes not only an Hpt domain but also trans-membrane and signaling domains. We suggest that the use of RcsD as the direct target of IgaA has allowed the development of a poised signaling complex, without impinging on structures in RcsC necessary for histidine

kinase activity. In addition, this branched signaling pathway may leave RcsC able to respond to other regulatory signals.

## Multipartite interactions of RcsD and IgaA regulate signaling: anchors and switches

Our analysis of the regions of RcsD and IgaA necessary for interaction and regulation suggests at least two points of contact between these proteins, each with different roles in regulation. A primary strong contact is proposed to be between the periplasmic domain of RcsD and the periplasmic domain of IgaA. The TM and periplasmic domains of RcsD (amino acids 1–383) are necessary but not sufficient for repression by IgaA; the periplasmic region of IgaA (aa 384–649) is similarly essential for IgaA function and drives the interaction of IgaA and RcsD in BACTH assays (Figs 2B and 4B). It seems likely that these periplasmic domains directly contact each other (Fig 6). Previous work by others has demonstrated that overproduction of the cytoplasmic loops of IgaA can support repression of the phosphorelay and allow depletion (but not deletion) of wild-type IgaA [10]. Presumably under these conditions, the loss of the periplasmic contacts is overcome by high levels of portions of IgaA mediating other repressive contacts. That leaves the question of whether the periplasmic contact we detect is part of the switch–is it responding to changes in signaling when all proteins are at their usual physiological levels? Certainly in some sense the periplasmic region of IgaA must be involved in signal transduction, since this is where RcsF talks to IgaA [5, 6, 10]. Whether RcsF and RcsD directly compete for binding to IgaA remains to be investigated.

However, our finding that a mutation in the cytoplasmic PAS-like domain of RcsD is sufficient to block signaling suggests that this critical interaction in the cytoplasmic domains serves as at least part of the switch. The cytoplasmic PAS-like domain of RcsD is necessary for regulation and for interaction with IgaA, measured both by BACTH assay and by titration of IgaA (Figs 2B and 3A). It seems likely that this cytoplasmic domain of RcsD contacts one or both cytoplasmic loops of IgaA. Our data favors a critical contact of the PAS-like domain of RcsD with the cytoplasmic loop 1 of IgaA (Fig 6). Both cytoplasmic loop 1 and cytoplasmic loop 2 of IgaA were necessary for RcsD to function properly (Fig 4C) in agreement with previous work [10]. We suggest that the interaction of cytoplasmic loop 1 and RcsD, in the region around T411, constitutes the regulatory switch for this system. Deletion of loop 1 is the least detrimental in terms of bacterial growth and signaling (Fig 4C), suggesting that the contacts outside cytoplasmic loop 1 are sufficient for enough IgaA repression of RcsD to support viability. Our model suggests that the additional repressive interaction in loop 1 is normally lost upon Rcs stimulus (in the presence of PMBN, for instance), and that the anchor contacts in the periplasm and possibly with IgaA loop 2 ensure that signaling is never so high that the cell dies. In the T411A mutant, this stimulus-sensitive contact becomes stronger, so that the system becomes uninducible (Fig 4A). This can be seen in the bacterial two-hybrid assay as some restored interaction in the absence of the IgaA periplasmic region (Fig 4B) and a decrease in mucoidy for the cytoplasmic loop 2 deletion (Fig 4C). We do not currently have any direct evidence that cytoplasmic loop 2 is contacting RcsD, but certainly deletion of this loop, like deletion of the periplasmic region, abrogates repression. This defect is partially suppressed by improving the interaction of IgaA with RcsD, via the T411A mutation (Fig 4C), strongly suggesting that T411 can operate in the absence of cytoplasmic loop 2.

In work by Collet and coworkers, overproduction of cytoplasmic loop 1 [10] was, by itself, capable of partially repressing an Rcs reporter in a strain depleted of IgaA; repression was to a similar extent to that seen with both cytoplasmic regions, supporting the critical role we

suggest here for the RcsD PAS-like domain contacts with IgaA cytoplasmic loop 1, highlighted by behavior of the T411A mutation.

## RcsD is an unusual phosphorelay protein

Phosphate flow in complex phosphorelays such as Rcs is from His (kinase domain) to Asp (response regulator domain of RcsC) to His (RcsD phosphotransfer protein) to Asp (RcsB response regulator domain). RcsD is a large inner membrane protein with many additional domains; its domain organization suggests that duplication of an ancestral protein may have given rise to RcsC and RcsD. Our results suggest critical roles for these additional regions of RcsD.

Consistent with RcsD's role as an anchor for IgaA, alignments suggest significant regions of conservation within the periplasmic domain of RcsD, apparently more so than the similarly sized RcsC periplasmic domain, which we show here is not critical for signaling (Fig 5A, S7 Fig). There is significant conservation as well in the truncated PAS domain, but less conservation in the inactive HATPase domain than in the active parallel RcsC domains (S7B Fig). Future work will be necessary to identify the periplasmic interaction points of RcsD with IgaA and to understand whether the RcsD HATPase domain plays any critical role in regulation.

## Alternative signaling pathways remain to be understood

The complexity of the Rcs phosphorelay provides opportunities for signals to regulate RcsB activity independently of the RcsF-IgaA-RcsD interaction network. Some transcription factors interact directly with RcsB, independent of its phosphorylation, to make heterodimers that regulate specific sets of genes (reviewed in [1]). In addition, there is evidence for activation of RcsB-dependent genes, dependent upon RcsC and RcsD, but independent of RcsF. For instance, overproduction of the DjlA DnaJ-like chaperone or mutation in *dsbA*, the gene encoding the periplasmic disulfide bond formation protein DsbA, both lead to RcsF-independent induction of the Rcs phosphorelay, possibly suggesting that alterations in protein folding may be the inducing event [1, 18, 27].

One other unexplored aspect of our work is the possible expression of low levels of the C-terminal domains of RcsD, to produce a short phosphotransfer protein that would not be subject to IgaA regulation. This is discussed in S1 Text, and remains to be explored in the future.

Overall, while a critical step in the best understood signaling pathway is clarified here, there is still much to learn about Rcs, other modes of signaling to the phosphorelay, and exactly how the IgaA/RcsD interactions modulate phosphate movement from RcsC through RcsD to RcsB. We do not yet know if RcsD acts as a monomer or a dimer, either with itself or RcsC, and whether its quartenary structure changes during signaling. Given the range of genes regulated by RcsB, and the importance of these genes for bacterial behavior, the complexity and potential flexibility in sensing and signal transduction in this system may not be surprising.

# Materials and methods

## Bacterial growth conditions and strain construction

Cells were grown in LB with appropriate antibiotics (ampicillin 100μg/ml, kanamycin 30–50μg/ml, chloramphenicol 10μg/ml for cat-sacB allele, usually listed here as cat or 25 μg/ml for chl$^R$ alleles, tetracycline 25μg/ml, gentamicin 10μg/ml, zeocin 50μg/ml); 1% glucose was added in some cases to reduce basal level expression of $P_{BAD}$ and $P_{Lac}$ promoters. For fluorescence/growth assays, strains were grown in MOPS minimal glucose or minimal glycerol (Teknova). Strains were constructed via recombineering and/or P1 transduction with selectable markers,

as outlined in S1 Table. Strains, plasmids, oligonucleotides and gBlocks used in this study are listed in S1, S2 and S3 Tables. Oligonucleotides and gBlocks were from IDT DNA, Coralville, IA.

For recombineering, cells carrying the chromosomal mini-λ Red system or the plasmid-borne Red system (pSIM27) were grown in LB, without or with Tetracycline respectively, at 32˚ C to an $OD_{600}$ of ~0.4–06. At mid-log, cultures were transferred to a water bath at 42˚ C to induce expression of the λ-Red system for 15 minutes and then immediately chilled in an ice-water slurry for 10 minutes prior to washing in sterile ice-cold water to make electrocompetent cells. 100 ng of ss oligo DNA or dsDNA (PCR product or gBlock) were used in the electroporation; 1 ml of LB or SOC was added for recovery before plating on selective plates [28]. Truncations and point mutations other than *rcsD*543, described in the strain table, were introduced in place of chromosomal copies of genes, leaving no marker or scar, by first deleting the gene of interest and inserting in its place the counter-selectable *ara-kan-kid* cassette from CAI_91. The cassette was then replaced with the desired allele, provided either as a PCR product or a gBlock. This cassette, a gift of C. Ranquet (BGene Genetics, Grenoble), expresses the Kid toxin under the control of an arabinose-inducible promoter. Cells carrying the *ara-kan-kid* counter-selectable marker cassette were grown with added 1% glucose in the media to repress. Counter-selection for removal of the *ara-kan-kid* cassette was done on LB-1% arabinose plates. All plasmid and chromosomal mutations were confirmed by sequencing using flanking primers.

Because many of the plasmids used here showed poor transformation using TSS, plasmids were generally introduced into cells using $CaCl_2$ transformation [21].

## DNA and strain manipulation and mutagenesis

Polymerase chain reactions were performed using Pfu Ultra II polymerase (Agilent) or Clontech Hifi polymerase (Takara). Primers used in this study are listed in S3 Table. PCR products were purified using column purification (Qiagen) according to the manufacturer's instructions. Gibson assemblies were performed using the Clontech In-Fusion HD Cloning Kit (Takara) and transformed into either Clontech Stellar Cells, NEB DH5alpha F'*lacI*Q or NEB Turbo cells containing LacI$^q$.

Alanine-scanning mutagenesis was carried out by SGI-DNA (San Diego, CA) using their BioXP system on pBAD24-RcsD (pEAW11). We ordered single mutants targeted at conserved residues within the cytoplasmic region of RcsD, from residue 326–683. The products from the company were first transformed into Stellar *E. coli* (Clontech), extracted and retransformed into EAW19, screening for fluorescence on minimal glucose-ampicillin agar plates, in comparison to a pRcsD$^+$ plasmid and the empty vector. Out of 35 mutants screened, ten had fluorescence levels comparable to the pRcsD$^+$ control and five of them were further studied (S5A Fig). Another 17 had higher fluorescence than either the pBAD vector or the pRcsD$^+$ control, but sequencing these isolates showed them to have picked up random mutations (stop codons) in addition to the designed ones and were not further studied. However, we note that the high levels of fluorescence for these isolates is best explained by assuming that they expressed, likely at a low level, the Hpt phosphotransfer domain, downstream from the observed stop codons.

## Bacterial Adenylate Cyclase Two-Hybrid Assay

In the bacterial adenylate cyclase two hybrid assay (BACTH), an adenylate cyclase mutant strain is used to assay for beta-galactosidase activity engendered when the T18 and T25 portions of adenylate cyclase are reconstituted, allowing cAMP/CRP to activate the *lac* operon. On their own, T18 and T25 will not form adenylate cyclase efficiently unless they are fused to

two interacting proteins [21]. Tags were C-terminal to avoid interference with protein insertion into the membrane. Primary data for the experiments in the Figures can be found in S1 Data.

Complementation assays were used to test the RcsD and RcsC fusion proteins for function. Plasmids expressing RcsD-T25 and RcsC-T25 were introduced into strains containing deletions for the appropriate gene; after transformation, the cells were transduced with P1 grown on strain NM357, containing *igaA*::chl[R], selecting for chloramphenicol resistance. In a strain deleted for *rcsD* or *rcsC*, the *igaA* deletion can be introduced by P1 transduction. However, the fusion plasmids blocked the ability of cells to be transduced with *igaA*::chl[R], consistent with them complementing their respective deletions (S2F Fig).

### *igaA* co-transduction frequencies

*bioH*/*igaA* co-transduction frequencies were used to determine which strains could support loss of IgaA. *bioH*, at 3544844 nt, is linked to *igaA* (position 3526469). The *bioH*::kan[R] mutant from the Keio collection [29] was introduced by P1 transduction into an *rcsD*541 *igaA*::chl[R] mutant (EAW17), selecting for kanamycin resistance and retention of chloramphenicol resistance (*igaA*::chl[R]), to create the donor strain EAW66 (*rcsD*541 *bioH*::kan[R] *igaA*::chl[R]). Because *rcsD* is inactive in this strain, it can tolerate loss of *igaA*. P1 transduction from this donor to recipient strains was carried out, selecting for Kanamycin Resistance and then screening 50–100 colonies for linkage to *igaA*::chl[R]. In *rcsB*, *rcsC* or *rcsD* null recipients, the co-transduction frequency was 78%. In a wild-type strain, the linkage dropped to zero, consistent with the known lethality of an *igaA* deletion [4, 5] (S4D Fig).

### Fluorescence assays

Fluorescence assays for Rcs activation were performed in 96 well plates in a Tecan Spark 10m or a Tecan Spark spectrophotometer. Strains carried a transcriptional fusion of mCherry, at the *ara* locus, to the promoter for sRNA RprA, as a reporter for Rcs pathway activation, referred to here as $P_{rprA}$-mCherry. Fluorescence of cells was measured in MOPS glucose minimal media (Teknova) unless otherwise stated. The pBAD24 plasmid was used for overexpression of RcsD fragments in strains expressing *araE* constitutively to ensure homogenous arabinose uptake [30]. For cells expressing proteins from pBAD, overnight cultures in MOPS minimal 0.2% glucose were washed with MOPS minimal glycerol to eliminate residual glucose, then diluted into fresh MOPS minimal glycerol media (.05% glucose, 0.5% glycerol) with 0.02% arabinose or 0.2% glucose as an uninduced control. Polymyxin B nonapeptide (PMBN; Sigma), a non-toxic polymyxin derivative, was used at 20 or 40 ug/ml, as indicated in figure legends, to induce the Rcs system. To check for Rcs induction by other known compounds, A22, an MreB inhibitor was used at 5ug/ml, and mecillinam was used at 0.3ug/ml.

Each combination of a strain and condition was performed in technical triplicate in the microtiter plate, with biological replicates performed on different days. Optical density and mCherry fluorescence were monitored every fifteen minutes for seven hours (S1A, S1B and S1C Fig). At the end of six hours, measurements of fluorescence at equivalent $OD_{600}$ values (0.4 +/- 0.03 after starting at $OD_{600}$ .03-.05) were converted to bar graphs of fold change of fluorescence with respect to the wild type strain. Some strains arrested growth early and never achieved 0.4 $OD_{600}$, and the $OD_{600}$ at 6 hours for those are noted on the graphs. Six hours marks the time when the wild type strain begins to transition to stationary phase, and ODs become less interpretable due to cell aggregation in the well bottom. Primary data for all Figures can be found in S1 Data.

## Western blots

For the chemiluminescent western blots in S2I and S5C Figs, plasmids transformed into NEB Turbo cells (NEB, Ipswich, MA) were used. Colonies were inoculated into LB media containing Amp (100 µg/ml) and grown for 3–4 hours at 37 C. At that point, IPTG was added to a final concentration of 1mM and cultures allowed to grow further. At 30, 60, and 120 min post-induction, a 1 ml aliquot was taken and TCA precipitated, washed in acetone and resuspended in sample buffer standardized to OD. 10 µl volumes were loaded and run on 4–12% NuPage gradient gels (Invitrogen, CA). Transfer was done in an iBlot2 onto nitrocellulose membranes as per manufacturer's specifications (Invitrogen, CA). After blocking in 5% milk, membranes were incubated in CyaA T18 mouse monoclonal primary antibody at 1:5000 (Santa Cruz Biotechnology, CA) overnight at 4 C, washed and incubated with an anti-mouse AP-conjugated secondary antibody (1:10,000) (Cell Signaling, MA) for 2 hrs. Development was with Novex CDP Star with Nitro Block II (Invitrogen, CA) according to manufacturer's instructions and exposed on a BioRad ChemiDoc MP imager (BioRad, CA).

For fluorescent western blots (S1E, S2G, S2H, S2J and S4A Figs), strains were grown in LB media to mid-log OD~ 0.4. A 1 ml aliquot was precipitated and washed in TCA-acetone as above. Primary rabbit polyclonal RcsD or RcsB antibodies were used at 1:5,000 (Covance, PA) while the loading control primary antibody was a mouse monoclonal anti-Ef-Tu at 1:10,000 (Hycult Biotech, PA). Secondary fluorescent antibodies anti-rabbit Starbright700 and anti-mouse DyLight800 were used at 1:5000 and 1:10,000 respectively (BioRad, CA). Imaging was done on a ChemiDoc MP.

## Supporting information

**S1 Fig. Measurement of Rcs activity by a fluorescent assay (relevant to Fig 1B).** S1A-C use a similar color code as Fig 1B, but with two shades of each color to indicate growth with or without PMBN; strains and treatments are shown with their color code. A. Growth curve of each strain +/- PMBN 20 µg/mL as shown in Fig 1B. Dotted lines represent an $OD_{600}$ of 0.4 (horizontal line) and a 360 min (6 hour) time point (vertical line), used as the standard measurements for fluorescent strains, unless otherwise indicated. Demonstrated in A is that stationary phase doesn't begin for any strain until close to or after $OD_{600}$ 0.8 under these growth conditions. Stationary phase always induces Rcs and can cause buildup of cells in well bottoms; therefore, measurements were not made past $OD_{600}$ 0.8 (solid horizontal line in panel A). Throughout the figures, if a strain has a growth defect that does not allow it to reach $OD_{600}$ 0.4 before the 360 min time point, it is noted with its actual $OD_{600}$ on the relevant bar graph legend. B. Relative fluorescent units (RFU) as a function of $OD_{600}$ for strains used in Fig 1B. The vertical dotted line represents the measurement point that is shown in the Fig 1B bar graph, $OD_{600}$ 0.4. These traces demonstrate the overall differences in Rcs activation of each strain. The effect of PMBN on the slope of each line can be seen clearly. For example, WT without PMBN (black) has a low slope throughout the graph, while WT + PMBN (gray) has a noticeably higher slope. The *rcsC* or *rcsD* mutants (blue and green respectively), have slight differences in RFU between treated and untreated conditions at each growth point; these differences do not dramatically affect the overall slope of the trace, indicating that small fluorescence differences here do not represent activation of Rcs as a whole. When a strain stops growing (for instance, as with WT+PMBN, gray line at $OD_{600}$ near 0.8) and the fluorescence continues to increase, the slope of the line becomes much sharper; we avoid using measurements in this range. C. Enlarged version of portion of S1B Fig. with only WT, *rcsF⁻* and *rcsB⁻* strains. Graph demonstrates that the point of divergence between the treated and untreated lines can be a useful proxy for detecting Rcs activation. True Rcs activation occurs in early growth points and is

consistent over the growth of the strain (see WT). This is also the case for an *rcsF* mutant (orange) which has a lower basal level of signal, but the PMBN-treated condition demonstrates a consistently higher slope, with no trace overlap after $OD_{600}$ of about 0.1. The *rcsB* deletion (red) gives a low slope with no reaction to PMBN, showing almost complete trace overlap. D. Different *rcsD* alleles have somewhat different behavior. *rcsD* is encoded upstream of *rcsB*, with the major promoters for *rcsB* inside the *rcsD* coding region [31]. This affects the way *rcsD* deletion alleles can be constructed. In addition, in both *Salmonella* and *E. coli*, some transcripts from the *rcsD* promoter may continue through to *rcsB*, though apparently at a much lower level [31, 32]. Four different chromosomal *rcsD* mutants, depicted in the gene schematics, were examined and were found to have modestly different effects on $P_{rprA}$::mCherry activity. *rcsD*543 contains a non-polar Kanamycin resistance cassette that is transcribed in the opposite direction to *rcsD*; the Kan cassette deletes everything from the RBS to 540bp inside the *rcsD* ORF. Our most commonly used mutant, *rcsD*541, is a markerless deletion that results from Flp recombinase removal of the Kan cassette from a different construct, but it has the exact same deletion boundaries as *rcsD*543, with a frt scar and no reverse promoter. Note that the basal level of expression of *rcsD*541 is somewhat higher than that for the other alleles; the reason for that is not currently clear. Strains shown: WT (EAW8), *rcsD*543 (EAW9), *rcsD*541 (EAW19), *rcsD* H842A (active site Hpt domain mutant, EAW57) and *rcsD*841* (two stop codons replace codons 842 and 843; EAW120). E. Western blots of RcsD in mutant strains and lack of polarity on RcsB. Left panel: Samples [1: Wild type (EAW8), 2: complete deletion of RcsD ORF with the kan^R AraC Kid cassette (EAW52)] were probed with polyclonal RcsD antibody. Right panel: Parallel detection of RcsD and RcsB to check RcsD alleles for RcsB polarity. Samples were blotted with both polyclonal RcsD antibody (both panels) and polyclonal RcsB antibody (right panel). Underlined constructs produce RcsD at expected molecular weight: 1) WT (EAW8), 2) *rcsD*543 (EAW9), 3) *rcsD*541 (EAW19), 4) RcsD H842A (EAW57), 5) *rcsD*841* (EAW120), 6) RcsD T411A (EAW121). The RcsD antibody can detect full length protein, but also detected a nonspecific band only slightly lower in molecular weight. In the right panel, *rcsD*841* was expected to make a truncated protein, but in a Western blot was found to have no identifiable protein in the correct size range. Although our inability to detect the protein may be due to the sensitivity of the antibody, we tentatively conclude that *rcsD*841* is probably a true RcsD null; the difference in expression with *rcsD*541 in S1D Fig is intriguing but unexplained. Further differences in these *rcsD* alleles were seen upon introduction of plasmids expressing some truncated RcsD derivatives (see S3B Fig, discussed in S1 Text). *rcsD* H842A produces a protein of the correct size, but has the same level of $P_{rprA}$-mCherry activation as deletion alleles 543 and 841*, as expected if it is devoid of phosphatase activity. As previously seen [13], *rcsD*541 and *rcsD*543 had no significant effect on RcsB levels, nor did *rcsD* H842A and *rcsD*841* (S1E Fig, right panel, RcsB band). F. An *ackA* mutant accumulates higher levels of acetyl phosphate, leading to phosphorylation of RcsB and thus activity of the $P_{rprA}$-mCherry reporter. *ackA* mutants were compared to *ackA*+ cells for expression of the $P_{rprA}$-mCherry reporter in a set of *rcsD* and *rcsC* mutants, grown in the absence of PMBN. The increase in reporter expression is modest (two-fold) in a strain wild-type for the Rcs phosphorelay in the absence of *ackA* (WT; black and gray bars, EAW122). The increase is fully dependent upon RcsB (right-hand brown bar, EAW126). The significantly higher activity in the *rcsC* and *rcsD* mutants is interpreted as a defect in dephosphorylation of RcsB~P. Thus, *rcsD*541, 841* and H842A (blue (EAW123), purple (EAW131) and green (EAW124) bars) all lose the ability to dephosphorylate RcsB, easily evident in an *ackA* background. Although both are apparently unable to fully dephosphorylate RcsB~P, a markerless whole-ORF *rcsC* deletion (EAW128; no RcsC receiver domain) and *rcsC* H479A (EAW129; intact RcsC receiver domain) appear to differ in their ability to perform the phosphatase reaction, consistent with

existing literature about the primacy of the receiver domain of RcsC in the dephosphorylation reaction [18]. All *ackA* mutants have a slight growth defect (right panel); *rcsD*841* is the most defective. For this strain, the sample was taken at $OD_{600}$ 0.24, possibly leading to an underestimate of its activity.
(TIF)

**S2 Fig. Interaction of IgaA and RcsD in a Bacterial Two-Hybrid Assay (relevant to Fig 2).** A. IgaA and RcsD interact well regardless of which tag is used on each. The interaction registers at least 1000 Miller units, while vector control experiments yield only 50, giving a 20-fold signal to noise ratio. Plasmids used: pEAW1 (IgaA-T18), pEAW2 (IgaA-T25), pEAW7 (RcsD-T25), pEAW8 (RcsD-T18). All error bars throughout the figures represent standard deviation. B. IgaA and RcsD interact robustly compared to control empty vectors, regardless of strain background. IgaA/RcsC interaction was below the limit of detection in all strains tested. Empty vector controls were performed in the WT background (BTH101), *rcsB*::Tn10 (EAW1), and *rcsC*::Tn10 (EAW2) and averaged to use as background. C. Interaction of RcsD and IgaA occur irrespective of strain background. Results from S2B merged with results from different experiments done in the *rcsF⁻* (EAW4) and *rcsD⁻* (EAW12) backgrounds. Each bar represents the relative IgaA/RcsD interaction measurement in the respective mutant host relative to the IgaA/RcsD interaction in wild type cells; this positive control is present for normalization in every assay of interaction of RcsD and IgaA wild type and mutants. D. RcsC interaction with IgaA or RcsD cannot be reliably detected irrespective of tag orientation. Left panel: IgaA/RcsC were fused in both orientations and tested in the BTH101 host. The dotted line at 200 Miller units represents approximately 4-fold over the background controls, the standard used in this work for a consistent, repeatable interaction determination. Note difference in beta-galactosidase values for even the strongest interaction here (150 Miller units) compared to the interaction of RcsD with IgaA (S2A Fig). Plasmids used: pEAW1 (IgaA-T18), pEAW6 (RcsC-T25), pEAW2 (IgaA-T25), and pEAW5 (RcsC-T18). V: vector, pUT18 for the T18 vector and pKNT25 for T25 vector. Right panel: RcsC-T18 (pEAW5) and RcsC-T25 (pEAW6) were tested for interactions with each other and with RcsD (RcsD-T25 (pEAW8) and RcsD-T18 (pEAW7)), but failed to give a positive interaction. A weak but positive interaction was detected in the same assays for RcsC$_{326-C}$-T18 (pEAW5s) with RcsD-T25, as also shown in S6A Fig. E. RcsC deleted for the periplasmic domain also does not interact with IgaA. Experiment is as for panels A-D. RcsC$_{\Delta45-314}$-T18 (pEAW5peri) was tested with RcsD-T25 (pEAW8), and IgaA-T25 (pEAW2). In parallel experiments, RcsD interaction with IgaA would yield at least 300 Miller units. F. RcsD-T25 and RcsC-T25 fusions are functional, as judged by complementation of chromosomal mutations for lethality in the absence of IgaA. When the *rcsC* strain EAW91 and *rcsD*541 strain EAW19 containing empty vector (pKT25) was transduced with P1 grown on NM357, a donor containing a chloramphenicol resistant *igaA* deletion allele (*igaA*::*chl*), many colonies resulted (left plate in each pair), because IgaA is only essential when the Rcs system is able to actively signal. When these strains contain RcsC-T25 (pEAW6) or RcsD-T25 (pEAW8) respectively, the Rcs signaling cascade is restored and deletion of *igaA* is no longer possible (right plate in each pair), demonstrating functionality of the RcsC-T25 and RcsD-T25 constructs. Rare colonies that do result on these plates are mucoid and/or mutant. G. Expression of RcsD-T18 Fusion proteins and detection by antibody to RcsD and to T18 CyaA. Western blot of RcsD-T18 fusion proteins, in a *cya+* strain (DH5 alpha *lacI*$^Q$), in which the fusion proteins are expressed whether or not they form detectable interactions. Transformed cells were grown in LB ampicillin (100 μg/ml) at 37°C to an OD600 of 0.3, and induced with 1 mM IPTG for one hour. Samples were taken and analyzed on parallel gels probed with either the anti-RcsD antibody, at 1:5000 dilution or anti-CyaA antibody, at

1:10,000 dilution. A comparison of these gels shows that, based on the anti-CyaA antibody detection, all of the fusions except that in lane 4 accumulate to similar levels, with somewhat more of the full-length RcsD (lane 7). However, the anti-RcsD antibody is significantly more effective in detecting proteins that contain the C-terminal Hpt domain (lanes 1, 4, 5, 7, and 8). Also apparent in some lanes are degradation products that may be lacking the C-terminal T18 tag. The host used is $rcsD^+$, and low levels of the chromosomally encoded RcsD (smaller than the wild-type RcsD-T18 in lane 1 because it lacks the T18 tag) can be detected as well (see lanes 2, 4–6). Plasmids present are Lane 1: pEAW7 (RcsD-T18), Lane 2: pEAW7alpha (RcsD$_{1-522}$-T18),Lane 3: pEAW7b (RcsD$_{1-683}$-T18), Lane 4: pEAW7c (RcsD$_{ABL-Hpt}$-T18), Lane 5: pEAW7d (RcsD$_{Hpt}$-T18), Lane 6: pEAW7m (RcsD$_{1-383}$-T18), Lane 7: pEAW7s (RcsD$_{326-C}$-T18), Lane 8 pEAW7peri (RcsD$_{\Delta45-304}$-T18). H. IgaA fusions with T18 are expressed. NEB turbo cells expressing IgaA derivatives were probed with the anti-CyaA antibody and EF-Tu. Lane 1: ladder, Lane 2: pEAW1 (IgaA-T18), Lane 3: pEAW1cyt1 (IgaA$_{\Delta36-181}$-T18), Lane 4: pEAW1cyt2 (IgaA$_{\Delta263-330}$-T18), Lane 5: pEAW1peri (IgaA$_{\Delta384-649}$-T18). I. RcsC-T18 fusion proteins are expressed. As for H, with plasmids expressing RcsC derivatives as well as RcsD$_{1-383}$: Lane 1: ladder, Lane 2: pEAW5 (RcsC-T18), Lane 3: pEAW5H (RcsC$_{H479A}$-T18), Lane 4: pEAW5s (RcsC$_{326-C}$-T18), and Lane 5: pEAW7m (RcsD$_{1-383}$-T18). Cells used in Lane 6 did not contain a T18 plasmid. J. Detection of RcsD-T25 derivatives with anti-RcsD antibody. Western for plasmids containing RcsD-T25 derivatives used in Fig 2C, grown in DH5alpha $lacI^Q$, induced for one hour with IPTG and probed with anti-RcsD antibody. The plasmids included here encode T25 derivatives of RcsD: Lane 1: RcsD (pEAW8); Lane 2: RcsD$_{\Delta45-304}$ (pEAW8peri); Lane 3: RcsD$_{1-461}$ (pEAWm2); Lane 4: RcsD$_{1-461,\Delta45-304}$ (pEAW8m2peri). The EF-Tu staining is shown in parallel because it runs at the same size as the RcsD$_{1-461,\Delta45-304}$ protein.

(TIF)

**S3 Fig. Analysis of RcsD function in signaling.** A. Signaling upon expression of RcsD N-terminal fragments. As for Fig 3A, with additional plasmids. RcsD C-terminal truncation constructs were expressed from arabinose-inducible plasmids in a WT (EAW8) and an $rcsD$541 (EAW19) host. The graphs of strain fluorescence (RFU) as a function of OD$_{600}$ for cells grown with arabinose are presented below their respective bar graphs. Constructs are color-coded: black: V, (pBAD24); blue: RcsD$^+$, (pEAW11); green: RcsD$_{1-461}$ (pEAW11m2); orange: RcsD$_{1-522}$ (pEAW11alpha); red: RcsD$_{1-683}$ (pEAW11b). Note that a change in slope on the fluorescence/ OD$_{600}$ graph demonstrates some level of P$_{rprA}$-mCherry activation, and that the orange (RcsD$_{1-522}$) and red (RcsD$_{1-683}$) slopes are very different from other slopes in the $rcsD$541 strain. Cell lysis can be seen as a reduction in OD$_{600}$ resulting in a leftward shift in the line (see orange and green lines in $rcsD$541 host). Note that, in spite of lysis for RcsD$_{1-461}$ in $rcsD$541, greater fluorescence did not result, compared to the vector control in the same time period. Therefore, lysis does not automatically increase P$_{rprA}$-mCherry fluorescence. Highest RFU with vector shown by horizontal dotted line, for comparison with experimental curves. This data and results in S3B are further discussed in S1 Text. B. Activity of RcsD plasmids in different $rcsD$ mutants. Based on the unexpected signal from plasmids lacking the Hpt domain in $rcsD$541 (S3A Fig), three additional $rcsD$ alleles were tested with RcsD C-terminal truncation plasmids. Fluorescence as a function of OD$_{600}$ is shown for cells grown with arabinose, as in S3A Fig, but in strains carrying the four different chromosomal $rcsD$ alleles, $rcsD$541 (EAW19, repeated from S3A Fig), $rcsD$543 (EAW9), $rcsD$H842A (EAW57) and $rcsD$841* (two stop codons at residue 841, EAW120), as previously studied without plasmids in S1D Fig. Each $rcsD$ allele is shown as an inset below the Fluorescence/ OD$_{600}$ trace for that strain. Plasmids are color-coded as in S3A Fig. Highest RFU with vector shown by horizontal dotted line.

$RcsD_{1-522}$ and $RcsD_{1-683}$ achieve higher slopes and/or final RFU values than the vector control in the $rcsD541$ and $rcsD543$ strains. The same is not true in backgrounds containing a disrupted Hpt domain ($rcsD$H842A or $rcsD$841*). How activation is occurring in $rcsD541$ and $rcsD543$ is unexplained, but it apparently requires the presence of the RcsD Hpt domain in the chromosome (see S1 Text). C. Overexpression of RcsD C-terminal truncations cannot activate in the absence of RcsB. These data demonstrate that activation of $P_{rprA}$-mCherry by RcsD plasmids is Rcs pathway specific. Assays and color-coding are as in S3A Fig, but in an $rcsB$::kan strain (EAW31). Shown here (L to R) are a bar graph with $rcsB$ RFU compared to WT, a bar graph ($OD_{600}$ 0.4 or final $OD_{600}$ value at 6 hours) where the RFU values for each construct are compared, and a graph of relative fluorescence units as a function of $OD_{600}$. There are no significant differences in slope or final RFU value, and the RFU values are the same as the background levels of $P_{rprA}$-mCherry expression in strains not expressing RcsB. D. Overexpression of RcsD constructs containing the Hpt domain depend on RcsC for high unregulated activation and mucoidy. In the left bar graph, RcsD on a plasmid was compared to empty vector in WT, $\Delta rcsC$ (EAW91) and $\Delta rcsC$ $rcsD541$ (EAW93) strains. Although signal increases in both $rcsC$ deletion backgrounds, the activation of the phosphorelay is not sufficient to lead to mucoidy in these strains. The threshold for mucoidy is closer to twelve-fold over wildtype; these strains approach seven-fold. In the right bar graph, color-coding is as in Fig 3B, with colors lighter with arabinose for each pair; all plasmids are in the $\Delta rcsC$ $rcsD541$ strain (EAW93). While the basis for higher reporter expression here is not known, the results may suggest a role for RcsD in mediating phosphate transfer to RcsB from sources other than RcsC.
(TIF)

**S4 Fig. Effects of RcsD Truncations on the ability to tolerate loss of IgaA.** A. Western blot for expression of pRcsD truncated proteins from the pBAD promoter tested in Fig 3C. Cells carrying pBAD-RcsD plasmids were grown in LB ampicillin (100μg/ml) with 1% glucose to an $OD_{600}$ of 0.3, washed and resuspended in LB ampicillin with 0.02% arabinose and allowed to grow for 2 hr before sample collection. Samples were run on a 4–12% gradient gel in MOPS buffer for 1 hr and probed with anti-RcsD antibody. Lane 1: $RcsD_{\Delta45-304}$, 632 aa (pEAW11 peri); Lane 2: $RcsD_{1-461}$, 458 aa (pEAW11m2); Lane 3: RcsD, 890 aa (pEAW11); Lane 4: $RcsD_{1-461, \Delta45-304}$, 174 aa (indicated with arrow). Note that the antibody preferentially detects RcsD derivatives carrying the Hpt domain, not present on the proteins in lanes 2 and 4 (see S2G Fig) B. Growth of the strains used in Fig 3C. Cell growth in glucose in the presence or absence of PMBN (left panel) or in the presence of arabinose, but in the presence or absence of PMBN (right panel) are shown. RFU units for cells that had grown to $OD_{600}$ of O.3, shown with a horizontal line on the graphs, were used to create the graph in Fig 3C. C. Signal activation and PMBN response for chromosomal mutants of RcsD. $rcsD$ alleles were introduced into the chromosomal $rcsD$ locus to create: $rcsD_{326-C}$ (EAW53), $rcsD_{\Delta48-304}$ (EAW106) and $rcsD_{686-C}$ (EAW108), grown and assayed as in Fig 1B. $rcsD_{792-C}$ could not be introduced without deleting promoters for RcsB, so that construct was not made. These alleles performed as their plasmid counterparts did, with the longer constructs roughly equivalent in their high signal and slow growth and the $rcsD_{686-C}$ allele appearing less efficient at passing signal to RcsB. Only the $rcsD_{686-C}$ allele can tolerate an igaA deletion (S4D Fig). D. Co-transduction of $igaA$::$chl^R$ with $bioH$::kan as an assay of Rcs function. Schematic shows $igaA$::$chl^R$ cotransduction frequency experiment using linked $bioH$::kan. The $bioH$::kan $igaA$::$chl^R$ P1 donor (EAW66) was constructed in an $rcsD541$ mutant. The table lists frequency of $igaA$::$chl^R$ cotransduction into various $rcs$ mutants, all isogenic derivatives of the $rcs^+$ strain EAW8, and also notes the phenotype of transductants. In some cases, as noted, while chloramphenicol resistant transductants were isolated, the resulting strains were unstable, and, based on their appearance and

fluorescence, likely rapidly accumulated secondary mutations.
(TIF)

**S5 Fig. Mutation in RcsD cytoplasmic region blocks signaling.** A. Plasmid-borne alleles in the RcsD cytoplasmic domain that retain phosphatase function. Strain EAW19 (*rcsD*541) with mutant derivatives of pBAD24-RcsD (pEAW11) were grown in MOPS glucose or MOPS glucose with PMBN, all with ampicillin and without arabinose induction and assayed for $P_{rprA}$-mCherry expression during growth. *rcsD541* has a higher signal than wild type and can be complemented with WT RcsD on a plasmid (compare lanes 1 and 3). The RcsD$^+$ construct responds to PMBN, unlike empty vector (compare lanes 1 to 2 and 3 to 4). Plasmids encoding *rcsD* alanine mutations in the cytoplasmic domains were screened for those that complemented *rcsD*541, reducing the basal level of expression; these were then assayed with and without PMBN. Of the 5 alleles shown here, 4 were inducible with PMBN. However, although it complements *rcsD541*, significantly lowering $P_{rprA}$-mCherry signal, interpreted as evidence of phosphatase activity, expression of the *rcsD*T411A point mutant was not induced in response to PMBN. B. BACTH IgaA loop deletion interactions with RcsD WT vs RcsD T411A. Experiment is as in Fig 4B, but showing additional controls. The IgaA+RcsD constructs gave β-galactosidase levels greater than thirty-fold over the single construct (background) controls. T18 derivatives carrying IgaA cytoplasmic loop one deletion (Δ36–181, cyt loop 1; pEAW1cyt1), IgaA cytoplasmic loop two deletion (Δ 263–330, cyt loop 2; pEAW1cyt2), IgaA periplasmic loop deletion (Δ 384–649, peri; pEAW1peri) were tested with RcsD-T25 WT (pEAW8) or RcsD T411A (pEAW8T); all were assayed in BTH101. C. IgaA point mutations in the periplasmic loop. Schematic showing point mutations surrounding IgaA L643P, a mutant of IgaA defective in Rcs negative regulation. In a western blot using the anti-T18 Cya antibody, the level of the T18-IgaA fusion protein was similar for L643P and wild type IgaA, ruling out protein instability as the explanation for its loss of function. EF-Tu was used as a loading control. Plasmids used: pEAW1, pEAW1L. D. Test of IgaA periplasmic loop mutations interaction with RcsD. Wild type IgaA interaction with RcsD in the BACTH system was set to one and compared to IgaA point mutant interactions with RcsD. IgaA L643P was deficient for this interaction, but L643A was significantly better and surrounding mutants were nearly WT for RcsD interaction. IgaA-T18 plasmids tested (L to R): pEAW1, pEAW1L, pEAW1LA, pEAW1D, pEAW1N, pEAW1H. RcsD-T25 plasmid: pEAW8. E. Effect of chromosomally expressed IgaA periplasmic loop mutations. When inserted into the chromosome in place of the wild-type *igaA* gene, only IgaA L643P produced Rcs dysregulation; the other mutants were wild-type for Rcs negative regulation and response to PMBN. Strains present (L to R): EAW8, EAW111, EAW112, EAW109, EAW110, EAW113.
(TIF)

**S6 Fig. The RcsC periplasmic loop is dispensable for signaling.** A: Bacterial two-hybrid assay of interaction of cytoplasmic portion of RcsC with RcsD but not IgaA. Plasmids used: IgaA-T25 (pEAW2), RcsD-T25 (pEAW8), and RcsC sol (RcsC$_{C326-C-}$T18, pEAW5s). B: Assays of RcsC periplasmic deletions and chimeric RcsC. RcsC periplasmic deletions perform differently when exposed to PMBN, depending on the linker length between transmembrane domains and the identity of those transmembrane domains. Strains present include (L to R) EAW8 (WT), EAW31 (*rcsB*::kan)), EAW61(*rcsC*$_{Δ45–314}$), EAW69 (*rcsC*$_{Δ47–314}$), EAW70 (*rcsC*$_{Δ48–314}$), EAW71 (*rcsC*$_{Δ49–314}$), and EAW72 (*rcsC*$_{MalF}$).
(TIF)

**S7 Fig. : Conservation in RcsC and RcsD.** A. Sequence alignments of 251 RcsC and RcsD homologs in Enterobacterales demonstrate differing regions of amino acid conservation.

Protein sequences were manually collected from 251 Enterobacterales species using NCBI Taxonomy Browser to determine genera for inclusion. Each genus in Enterobacterales was checked for species containing annotated adjacent RcsC and RcsD ORFs on their genomes using NCBI Protein and Nucleotide; species containing full sequences of the region were selected for inclusion in the alignment. RcsC and RcsD alignments were performed using ClustalW and the sequence logos were automatically generated in Geneious. High amino acid conservation is demonstrated by high lines in the logo and dark regions in the line below the logo, which represents a consensus sequence. Transmembrane regions are marked by red rectangles, domain regions are marked by black arrows. B. Expanded alignment for the Transmembrane and initial cytoplasmic regions of RcsC and RcsD. The location of T411A in the PAS-like domain of RcsD is shown with an arrow.
(TIF)

**S1 Data. These files contain the primary data used to generate graphs presented in the Figures and Supplementary Figures.** Each excel or Prism file is named to match the appropriate figure or figures, with the date on which the data was generated, and in some cases with additional information.
(ZIP)

**S1 Text. A possible role for the RcsD Hpt domain in the unexpected activation of the phosphorelay with certain RcsD mutant proteins.**
(DOCX)

**S1 Table. Strains used. Strain name, relevant genotype and method of construction or reference, if previously published, are shown.** Details of construction are provided in Materials and Methods.
(XLSX)

**S2 Table. Plasmids used.** Details of construction are provided in Materials and Methods.
(XLSX)

**S3 Table. Primers and gBlocks used.** "Description" column provides information on the way in which the primer or gBlock was used. Non-underlined sequence followed by "*" denotes 5' overlap with another sequence, vector or another insert for use in In-Fusion reaction (Gibson assembly). Underlining indicates annealing region from which Tm is calculated.
(XLSX)

## Acknowledgments

We thank Caroline Ranquet for the generous gift of the kan-araC-Kid cassette. We thank S. Buchanan and members of her laboratory as well as members of the LMB for discussion throughout this work. We thank A. Petchiappan, M. Maurizi and S. Stibitz for comments on the manuscript.

## Author Contributions

**Conceptualization:** Erin A. Wall, Nadim Majdalani, Susan Gottesman.

**Formal analysis:** Erin A. Wall, Nadim Majdalani, Susan Gottesman.

**Funding acquisition:** Erin A. Wall, Susan Gottesman.

**Investigation:** Erin A. Wall, Nadim Majdalani, Susan Gottesman.

**Methodology:** Erin A. Wall, Nadim Majdalani, Susan Gottesman.

**Project administration:** Susan Gottesman.

**Supervision:** Susan Gottesman.

**Visualization:** Erin A. Wall, Nadim Majdalani.

**Writing – original draft:** Erin A. Wall, Nadim Majdalani, Susan Gottesman.

**Writing – review & editing:** Erin A. Wall, Nadim Majdalani, Susan Gottesman.

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
