## [Decision Letter · Decision Letter 0]

11 Feb 2020

Dear Dr Gottesman,

Thank you very much for submitting your Research Article entitled 'Negative regulation of the Rcs phosphorelay via IgaA contact with the RcsD phosphotransfer protein' to PLOS Genetics. Your manuscript was fully evaluated at the editorial level and by independent peer reviewers. The reviewers appreciated the attention to an important topic but identified some aspects of the manuscript that should be improved.

We therefore ask you to modify the manuscript according to the review recommendations before we can consider your manuscript for acceptance. Your revisions should address the specific points made by each reviewer.

[LINK]

Yours sincerely,

Sean Crosson

Associate Editor

PLOS Genetics

Lotte Søgaard-Andersen

Section Editor: Prokaryotic Genetics

PLOS Genetics

Reviewer's Responses to Questions

**Comments to the Authors:**

Reviewer #1: The Regulator of Capsule Synthesis (Rcs) is one of the most complex signaling pathways in bacteria. It is structured around the RcsCDB His-Ass phosphorelay, but is controlled by upstream protein regulators, the sensory component RcsF and the phosphorelay inhibitor IgaA. The regulatory mechanism of Rcs is poorly understood. Over-activation of the stress response is lethal and, for this reason, igaA is essential unless any of the downstream components RcsCDB components are inactivated. While this genetic analysis cannot determine the direct target of the IgaA inhibition, the hypothesis was put forward that IgaA inhibits the RcsC kinase, as it initiates the phosphorelay. This hypothesis was never directly tested but propagated over the years so many times that the recent Rcs literature states it as a fact and not a hypothesis in the introduction and discussion sections. In this manuscript, Wall et al used protein interaction studies and extensive genetic analysis to demonstrate that the phosphotransfer protein RcsD, and not RcsC, is the direct target for IgaA inhibition. This study is significant because by experimentally challenging this old assumption, it revisits the regulatory mechanism of a major envelope stress response in enteric bacteria. It will be general interest to anyone who studies regulatory pathways.

My major comments aimed to strengthen and clarify some of the aspects of a proposed new signaling model and to improve flow and readability.

1. Lines 494-499: Authors suggest that the “strong periplasmic interaction provides an anchor for interaction with RcsD, but likely not the region in which signal is sensed”. In my opinion, the observation that overexpression of IgaA missing the periplasmic loop is not sufficient to make a statement about the sensing mechanism. It is common that a decrease in affinity can be overcome by an increase in concentration. I am surprised that there is no discussion about RcsF interaction with IgaA periplasmic loops in this context. In the simplest scenario, RcsF could compete with RcsD for IgaA binding and in that way constitute the “sensing” mechanism.

a) Does RcsF overexpression or inner membrane-mislocalized RcsF affect the IgaA/RcsD binding in BTH?

b) Is the strain overexpressing IgaA without the periplasmic loop competent for sensing? It can be constitutively OFF.

2. Authors suggest that the RcsD “PAS” domain plays an important role in the IgaA-mediated regulation, mainly because of the expression of the RcsD N-terminal domain (periplasmic loops and transmembrane helixes) is not sufficient for a full display of phenotypes.

a) Is it possible that the cytoplasmic domains of RcsD play an indirect role by facilitating dimerization and thereby affecting the quaternary structure of the RcsD periplasmic domain? Many PAS domains also act as dimers.

b) The phenotype of the “PAS domain only” construct was not tested but is important for the overall conclusion of the paper. Is there a detectable interaction with IgaA in BTH or titration experiments? While there is no detectable interaction with “326” construct, as authors discussed, other cytoplasmic domains may be inhibitory.

3. What is a specific model for how IgaA/RcsD interaction inhibits phosphorelay? Do authors think that RcsD is physically titrated away from RcsC or all proteins form a stable complex that undergoes the conformational change? Does RcsC/D interactions in BTH change in the absence of IgaA (in rcsB background)? It should be easy to test since authors have all of the constructs on hands. Regardless of the outcome, I think it will be valuable information for future mechanistic studies.

4. Authors performed very extensive analysis to test for IgaA/RcsD/RcsC interactions using bacterial-two hybrid (BTH), growth, signaling phenotype with many different alleles in various genetic backgrounds. Following these multiple phenotypes is quite challenging throughout the manuscript and requires constant flipping back and forth between text, figures, and extensive supplementary material. I suggest that adding a summary of all phenotypes for each of the mutant alleles in a table format would be helpful for readers. It will also enable authors to reference this table in the discussion section as opposed to referring back to the original multi-panel figures.

5. Likewise, there are many genetic constructs with less conclusive phenotypes that are extensively discussed in the main text with references to the supplementary figures. These mutants will be a great asset for those working directly on Rcs; but, in my opinion, they often take away from the main message of the paper. While I let authors to decide which are the most critical to keep in the main text, I would encourage them to move as many details to the corresponding supplemental figure legends in order to preserve the flow of the manuscript.

Other specific comments:

Line 84: “Bar graphs of fluorescence at OD600 0.4.” is a sentence fragment.

Lines: 139-144 and Fig. S2: What is the nature of a 4-fold cutoff?; it seems arbitrary. While small, there is a detectable interaction between IgaA and RcsC that seems to be RcsD-dependent and it looks significant based on the error bars.

Line 169 (entire section). Are levels RcsD comparable? There is no associated immunoblot.

Lines 221-224 and Fig. S1E: Are there any smaller fragments in the RcsD immunoblot?

Lines 388-392: Negative results here are inconclusive about the importance of the L643 for direct interactions. Ala substitution may not have a phenotype because of similar properties. I would just rephrase that L643P may either directly or indirectly affect binding.

Lines 473: “A22 and Mecillinam, peptidoglycan disruptors”. Please rephrase as it implies physical disruption, which is not the case.

Lines 530-534 and figure S6: I think the difference in conservation is very intriguing; is it possible to give some quantitative value to this observation? The figure needs to be properly explained. Also, I think it’s important to use rcsC and rcsD sequences from the same species representative to avoid the sequence bias. Why was the different number of RcsC and RcsD sequences used?

Figure 1A: This figure is reused from the recent review article by the same authors. Please verify the permissions for that.

Figure 4C: I don’t think it’s essential and can be moved to the supplement.

Reviewer #2: The Rcs phosphorelay is a particularly complex stress signaling system that senses and responds to envelope damage. Stress conditions change the interaction between the outer membrane stress sensor lipoprotein RcsF and the essential inner membrane protein IgaA, which in turn impacts the interaction between IgaA and the downstream components of the phosphorelay, turning Rcs on. How IgaA interacts with the phosphorelay and controls its activity was unknown. In this article, Wall and coauthors use an elegant combination of in vivo interaction assays (bacterial adenylate cyclase two hybrid assay – BACTH), genetic analysis, mutagenesis, phenotypic experiments and reporter assay experiments to investigate the interaction between IgaA and the two other inner membrane components of Rcs, the histidine kinase RcsC and the phosphotransfer protein RcsD. They found that, in contrast to what was generally assumed, IgaA regulates Rcs by interacting with RcsD and not with RcsC, although an IgaA-RcsC interaction cannot be ruled out. Using RcsD and IgaA fragments, they also dissected the interaction between the different domains of these two proteins. In sum, their results reveal that the periplasmic domain of IgaA functions as an anchor for the periplasmic domain of RcsD and that the periplasmic contact between these two proteins is necessary for repression. Crucial, although weaker, interactions also take place in the cytoplasm; these contacts, in particular with loop 1 of IgaA, are necessary for the regulation of Rcs. Additional experiments aiming at understanding the RcsD-RcsC interaction are also reported.

Overall, the paper is interesting and provides an important and novel contribution to the understanding of the particularly complex Rcs system. An impressive amount of data is presented: these data convincingly support the proposed model that the IgaA-RcsD interaction is crucial in controlling Rcs activity, which is a major step forward. The data are adequately discussed and thoroughly interpreted.

Here are my comments:

1-the paper, although well and nicely written, is extremely long, in particular the results section; this section describes key data, which are presented in the main manuscript, and that are central to the message of the paper. However, less important experiments, whose results are presented in the supplementary material, are also described at length. As a result, the reader loses focus and gets distracted, which makes the reading of the manuscript more challenging. I would suggest moving some of the description of the less crucial experiments to the supplementary material, to help the reader understanding the key message of the paper.

2. the results of the BACTH assays are convincing when an interaction is detected. However, the absence of signal does not provide an evidence that the tested partners do not interact. This is particularly true when it comes to the RcsC-IgaA potential interaction. To that respect, an important control is missing: the authors should test whether they can “detect” the RcsC-RcsC dimer , which would indicate that the RcsC constructs are working in the BACTH assay.

Reviewer #3: In this manuscript, the authors investigate the interactions between the Rcs signaling proteins which control an envelope stress response in Gram negative enteric bacteria. The provide evidence that the “brake” protein IgaA interacts with the membrane bound histidine phosphotransfer protein RcsD, but not the histidine kinase signaling initiator RcsC. They localize this region, using the BACTH assay to a putative PAS motif in the cytoplasm, but not containing the histidine phosphotransfer domain. They go on to show that RcsD constructs interacting with IgaA also have the ability to activate the wild-type Rcs pathway when expressed in trans, suggesting they have the ability to titrate the negative regulator IgaA away from the wild-type signaling complex. The authors go on to isolate a mutant in the RcsD PAS domain that both dampens signaling in response to inducing cues and appears to strengthen IgaA interaction through cytoplasmic domain 1 of IgaA. The RcsD PAS domain mutant is capable of diminishing Rcs signaling in strains containing IgaA cytoplasmic loop 1, further supporting this interaction. Finally, the authors go on to demonstrate that the periplasmic loop of RcsC is not involved in signaling in response to a number of envelope stresses, and suggest that RcsD interaction with IgaA is instead the point of signal reception in these cases.

This is a dense paper with a wealth of genetic data that add to our understanding of how the Rcs signaling cascade works. None-the-less, it was somewhat difficult to follow the thread of the story due to the sheer amount of data presented and the fact that it was split between the manuscript itself and the supplemental data. The genetic data supports the conclusions regarding RcsD:IgaA interaction, but it would be nice to have data demonstrating the physical interactions to validate the genetics. There are some inconsistencies in the data that the authors do a good job trying to address, but one also wonders whether these inconsistencies reflect unknown aspects of interactions between RcsD and RcsC or other proteins. This possibility is not really addressed.

Other Comments:

1. Line 84 – there is some sort of typo in this line. The sentence “Bar graphs of fluorescence at OD600 0.4” doesn’t fit.

2. Fig. 2B – it is a bit perplexing that the authors argue for an interaction between the periplasmic domain of RcsD and IgaA based on this data, since the construct lacking the cytoplasmic domain is one of the weakest interactors in this assay.

3. Fig. 3A – why does over-expression of WT RcsD not have a stimulatory effect? This protein binds to IgaA as demonstrated in the BACTH, and one would expect that over-expression should titrate out IgaA and lead to a similar stimulatory effect as the other constructs that interact with IgaA?

4. It is unclear what the experiments in Fig. 3B add. The negative data in the wild-type are consistent with previous conclusions that a loss of interaction with IgaA is correlated with a loss of the ability to interfere with wild-type signaling, but the data in the right panel only serve to underscore that in the presence of a mutant (very lowly expressed?) version of RcsD, the Hpt domain in these constructs is capable of moving phosphate through the signaling pathway. This data is a bit distracting from the main points the authors try to make regarding IgaA:RcsD interaction.

5. Line 176 “… expressed AT levels….”

6. Lines 249-251: The authors state that “Finally, because this signaling is not seen in an rcsD+ strain, signaling by these fragments of RcsD is recessive to the full-length protein.” Do they interpret this to mean that the RcsD derivatives do not interact with RcsC as well as WT?

7. Lines 313-315: The authors state that “Thus, while rcsD326-C was negative in the bacterial two hybrid interaction with IgaA (Fig 2B), the continued dependence on IgaA for viability is consistent with it retaining a critical contact with IgaA”. Could some of the inconsistencies between the interaction with IgaA observed in the BACTH and the genetic/complementation results also be due to altered interactions between RcsD and RcsC, which are not investigated here?

8. Ruling out an RcsC:IgaA interaction based solely on an artificial BACTH assay seems premature?

9. There is a lot of text at the beginning of the paper and in the supplementary data devoted to discussing the development of an rprA-mCherry reporter, which appears to behave very similarly to other previously published rprA reporter genes. Since the manuscript is quite long, the authors might consider condensing the discussion of the data presented around this topic.

10. A nice further validation of the authors’ model for IgaA:RcsD contacts would be to look at the impact of both the T411A and periplasmic loop mutations together. The model presented here would suggest that these mutations would be additive.

**Have all data underlying the figures and results presented in the manuscript been provided?**

Reviewer #1: Yes

Reviewer #2: Yes

Reviewer #3: Yes

PLOS authors have the option to publish the peer review history of their article (what does this mean?). If published, this will include your full peer review and any attached files.

Reviewer #1: No

Reviewer #2: No

Reviewer #3: No

---

## [Decision Letter · Decision Letter 1]

10 Jun 2020

Dear Dr Gottesman,

We are pleased to inform you that your manuscript entitled "IgaA negatively regulates the Rcs Phosphorelay via contact with the RcsD Phosphotransfer Protein" has been editorially accepted for publication in PLOS Genetics. Congratulations!

Yours sincerely,

Sean Crosson

Associate Editor

PLOS Genetics

Lotte Søgaard-Andersen

Section Editor: Prokaryotic Genetics

PLOS Genetics

Comments from the reviewers (if applicable):

Reviewer's Responses to Questions

**Comments to the Authors:**

Reviewer #1: This is a revised manuscript by Wall et al on regulation of the complex Rcs stress response. The authors did a great job addressing my concerns and clarifying their model. The new data on phenotypes of the membrane-tethered “PAS only” RcsD construct helped to strengthen the original conclusion. Major restructuring and rewriting of results section and supplement improved readability. Overall, it is a greatly improved manuscript that will be of interest to anyone working on regulatory systems. I don’t have any further comments.

Reviewer #2: I think that the authors have done an excellent job in revising the paper: new experiments have been done to address the comments and the organisation of the manuscript has been much improved.

Reviewer #3: I thank the authors for their comprehensive response to my comments.

**Have all data underlying the figures and results presented in the manuscript been provided?**

Reviewer #1: Yes

Reviewer #2: Yes

Reviewer #3: Yes

PLOS authors have the option to publish the peer review history of their article (what does this mean?). If published, this will include your full peer review and any attached files.

Reviewer #1: No

Reviewer #2: No

Reviewer #3: No

**Data Deposition**

http://datadryad.org/submit?journalID=pgenetics&manu=PGENETICS-D-20-00042R1

**Press Queries**

---

## [Editor Report · Acceptance letter]

6 Jul 2020

PGENETICS-D-20-00042R1 

 IgaA negatively regulates the Rcs Phosphorelay via contact with the RcsD Phosphotransfer Protein 

Dear Dr Gottesman, 

We are pleased to inform you that your manuscript entitled " IgaA negatively regulates the Rcs Phosphorelay via contact with the RcsD Phosphotransfer Protein " has been formally accepted for publication in PLOS Genetics! Your manuscript is now with our production department and you will be notified of the publication date in due course.

With kind regards,

Matt Lyles

PLOS Genetics

On behalf of:
